# DYNAMICAL MODELING FOR REAL-TIME INFERENCE OF NONLINEAR LATENT FACTORS IN MULTISCALE NEURAL ACTIVITY

## ABSTRACT

Continuous real-time decoding of target variables from time-series data is needed for many applications across various domains including neuroscience. Further, these variables can be encoded across multiple time-series modalities such as discrete spiking activity and continuous field potentials that can have different timescales (i.e., sampling rates) and different probabilistic distributions, or can even be missing at some time-steps. Existing nonlinear models of multimodal neural activity do not support real-time decoding and do not address the different timescales or missing samples across modalities. Here, we develop a learning framework that can nonlinearly aggregate information across multiple time-series modalities with such distinct characteristics, while also enabling real-time decoding. This framework consists of 1) a multiscale encoder that nonlinearly fuses information after learning within-modality dynamics to handle different timescales and missing samples, 2) a multiscale dynamical backbone that extracts multimodal temporal dynamics and enables real-time decoding, and 3) modality-specific decoders to account for different probabilistic distributions across modalities. We further introduce smoothness regularization objectives on the learned dynamics to better decode smooth target variables such as behavioral variables and employ a dropout technique to increase the robustness for missing samples. We show that our model can aggregate information across modalities to improve target variable decoding in simulations and in a real multiscale brain dataset. Further, our method outperforms prior linear and nonlinear multimodal models[1].

## 1 INTRODUCTION

Many engineering and science applications need to infer target variables that are encoded in multiple time-series modalities and do so in real-time (i.e., causally). An important example of such an application arises in neuroscience for the inference of cognitive or behavioral variables from multimodal neural time-series data. Accurate inference of these variables requires developing nonlinear dynamical models of the multimodal time-series that can, at each time-step, aggregate information across modalities in real-time. Further, a natural challenge in developing these models arises when modalities can be missing at some time-steps due to measurement failures or interruptions (Burger et al., 2018; Li & Marlin, 2020; Berger et al., 2020) or due to different timescales (i.e., sampling rates) across modalities (Hsieh et al., 2018; Lu et al., 2021). Therefore, designing a dynamical model of multimodal time-series data in real-time applications requires enabling temporal and multimodal data fusion, performing real-time inference, and addressing different timescales and/or missing samples across modalities. Prior nonlinear dynamical models have not addressed all these challenges. To that end, we develop a new nonlinear dynamical model of multimodal time-series that provides all these capabilities. Our model can perform multimodal data fusion and account for different timescales and missing samples, while also enabling real-time inference.

In neuroscience, real-time multimodal data fusion is important especially for the design of neurotechnologies such as brain-computer interfaces. It has been shown that fusing neural modalities such as local field potentials (LFP) and spiking activity can improve the inference of arm movements

---

[1]Our codebase is available at `https://anonymous.4open.science/r/mrine`

(Stavisky et al., 2015; Abbaspourazad et al., 2021; Ahmadipour et al., 2023). More specifically, spiking activity is a binary-valued time-series that indicates the presence of action potential events from neurons and count processes such as Poisson are employed for modeling spiking activity. LFP activity is a continuous-valued modality that measures network-level neural processes and is typically modeled with a Gaussian distribution (Einevoll et al., 2013; Lu et al., 2021). Further, LFPs are often obtained at a slower timescale than spikes, which are typically measured at a millisecond timescale (Lu et al., 2021). We refer to multimodal data with different timescales as **multiscale data**. Thus, to fuse information across the spiking and LFP modalities and improve downstream behavior decoding tasks, their dynamics should be modeled by incorporating their cross-modality probabilistic and timescale differences.

Most of the existing dynamical modeling approaches in neuroscience focus on a single modality of neural activity. These models are either in linear or switching linear form (Macke et al., 2011; Koh et al., 2023; Linderman et al., 2017) or utilize deep learning approaches for nonlinear modeling (Abbaspourazad et al., 2023; Archer et al., 2015; Gao et al., 2016; Pandarinath et al., 2018). However, these models do not capture multimodal neural dynamics. Even though some approaches aim to model single-modal neural activity and behavior as multimodal signals (Sani et al., 2021; Hurwitz et al., 2021; Schneider et al., 2023), their latent factor inference is performed by processing single-modal neural signals. In contrast, (Rezaei et al., 2023) and (Coleman et al., 2011) propose linear multimodal frameworks for neural activity, but they do not handle different timescales. Motivated by this, recent dynamical models have been designed for multiscale neural dynamics while also enabling real-time inference and handling timescale differences, but they are in linear form and cannot capture nonlinearities (Abbaspourazad et al., 2021; Ahmadipour et al., 2023). There have also been important recent studies on nonlinear modeling of multimodal neural data (Kramer et al., 2022; Brenner et al., 2022; Gondur et al., 2023; Zhou & Wei, 2020), however, their latent factor inference is done non-causally over time and does not handle different timescales. Beyond neural time-series data, many approaches in other domains have been proposed to combine multiple modalities to improve performance in downstream task. However, they are not focused on addressing the challenge of different timescales and missing samples over time, and their applicability to modeling of Poisson and Gaussian distributed modalities encountered in neuroscience has not been investigated (see Section 4).

**Contributions** We develop a novel nonlinear dynamical modeling method that can nonlinearly aggregate information across multiple modalities with different distributions, distinct timescales, and/or missing samples over time, while supporting inference both in real-time and non-causally. To achieve these capabilities and enhance performance compared with baselines, we: **1)** design a multiscale encoder that performs nonlinear information fusion through neural networks after learning modality-specific linear dynamical models (LDMs) that account for timescale differences and missing samples in real-time by learning temporal dynamics (Section 2.2), **2)** impose smoothness priors on the latent dynamics via new smoothness regularization objectives that also prevent learning trivial identity neural network transformations (Section 2.3) and **3)** employ a technique termed *time-dropout* during model training to increase robustness to missing samples even further (Appendix A). We term this method **M**ultiscale **R**eal-time **I**nference of **N**onlinear **E**mbeddings (MRINE).

Through stochastic Lorenz attractor simulations and real nonhuman primate (NHP) spiking and LFP neural datasets, we show that MRINE infers latent factors that are more predictive of target variables – i.e., true latent factors for the simulations and behavioral arm/manipulandum movement variables for the NHP datasets. Further, we compare MRINE with various recent linear and nonlinear multimodal methods and show that MRINE outperforms all methods in behavior decoding for the NHP datasets.

## 2 METHODOLOGY

We assume that we observe discrete neural signals (e.g., spikes) $s_t \in \{0,1\}^{n_s}$ for $t \in \mathcal{T}$ where $\mathcal{T} = \{1, 2, \ldots, T\}$ and continuous neural signals (e.g., LFPs) $y_{t'} \in \mathbb{R}^{n_y}$ for $t' \in \mathcal{T}'$ where $\mathcal{T}' \subseteq \mathcal{T}$. Note that the two different sets $\mathcal{T}$ and $\mathcal{T}'$ allow for the timescale differences of $s_t$ and $y_{t'}$ via different time-indices. As shown in Figure 1 and expanded on below, we describe the neural processes generating $s_t$ and $y_{t'}$ through multiscale latent and embedding factors, which in turn can be extracted by nonlinearly aggregating information across these multiple neural modalities with a multiscale encoder.

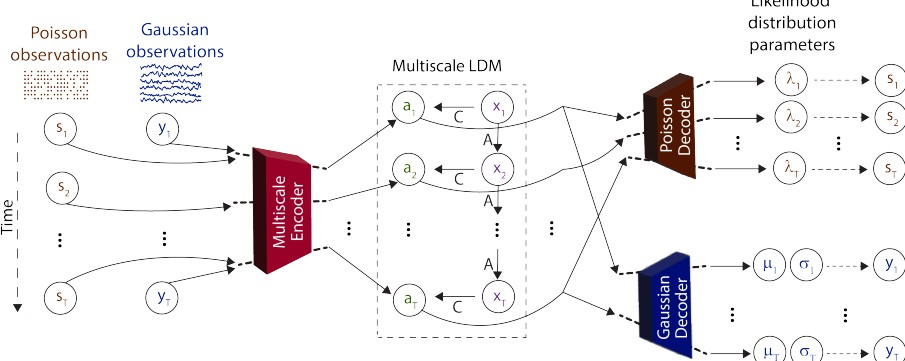

Figure 1: MRINE model architecture. Multiscale encoder nonlinearly (see Figure 2) extracts multiscale embedding factors ($a_t$) by fusing discrete Poisson and continuous Gaussian neural time-series in real-time (as indicated by thick dashed lines) while accounting for timescale differences and missing samples. Temporal dynamics on this multiscale embedding are then explained with a multiscale LDM whose states are the multiscale latent factors ($x_t$). Embedding factors reconstruct the distribution parameters of both the Poisson and the Gaussian modalities through the modality-specific decoders. As an example, when $y_2$ is missing, $a_2$ and $x_2$ are inferred by processing only $s_2$.

## 2.1 MODEL FORMULATION

The model architecture is shown in Figure 1 with its main components being a multiscale encoder for nonlinear information fusion, a multiscale LDM backbone, and decoders for each modality. We write the generative model as follows:

$$x_{t+1} = Ax_t + w_t \tag{1}$$

$$a_t = Cx_t + r_t \tag{2}$$

$$p_{\theta_s}(s_t \mid a_t) = \text{Poisson}(s_t; \lambda(a_t)) \tag{3}$$

$$p_{\theta_y}(y_{t'} \mid a_{t'}) = \mathcal{N}(y_{t'}; \mu(a_{t'}), \sigma). \tag{4}$$

Here, Eq. 1 and 2 form a multiscale LDM , and $t \in \mathcal{T}$ where $\mathcal{T} = \{1, 2, \ldots, T\}$ and $t' \in \mathcal{T}'$ where $\mathcal{T}' \subseteq \mathcal{T}$. In this LDM, $a_t \in \mathbb{R}^{n_a}$, termed multiscale embedding factors, are the observations. We obtain $a_t$ as the nonlinear aggregation of multimodal information through the multiscale encoder designed in Section 2.2. Further, $x_t \in \mathbb{R}^{n_x}$, termed multiscale latent factors, are the LDM state and model the linear dynamics in the nonlinear embedding space. Note that from this point onward, we will only use time-index $t$ (except observation models) for embedding and latent factors as $\mathcal{T}' \subseteq \mathcal{T}$. Correspondingly, $A \in \mathbb{R}^{n_x \times n_x}$ and $C \in \mathbb{R}^{n_a \times n_x}$ are the state transition and observation matrices in the multiscale LDM, respectively; $w_t \in \mathbb{R}^{n_x}$ and $r_t \in \mathbb{R}^{n_a}$ are zero-mean Gaussian dynamics and observation noises with covariance matrices $W \in \mathbb{R}^{n_x \times n_x}$ and $R \in \mathbb{R}^{n_a \times n_a}$.

The observations from different modalities $s_t$ and $y_{t'}$ are then generated from $a_t$ (or $a_{t'}$) as in Eq. 3 and 4, respectively, with the likelihood distributions denoted by $p_{\theta_s}(s_t \mid a_t)$ and $p_{\theta_y}(y_{t'} \mid a_{t'})$. Assuming that data modalities are conditionally independent given $a_t$ (or $a_{t'}$ for $y_{t'}$), we modeled the discrete spiking activity $s_t$ with a Poisson distribution and the continuous neural signals $y_{t'}$ with a Gaussian distribution, given that these distributions have shown success in modeling each of these modalities (Stavisky et al., 2015; Gao et al., 2016; Pandarinath et al., 2018; Abbaspourazad et al., 2021). The means of the corresponding distributions $\lambda(a_t) \in \mathbb{R}^{n_s}$ and $\mu(a_{t'}) \in \mathbb{R}^{n_y}$ are parametrized by neural networks with parameters $\theta_s$ and $\theta_y$. Practically, we observed that learning the variance of the Gaussian likelihood yielded suboptimal performance, thus we set it to a constant value, i.e., unit variance, as in previous works (Fraccaro et al., 2017; Henaff et al., 2018; Pong et al., 2020).

## 2.2 ENCODER DESIGN AND INFERRING MULTISCALE FACTORS

To infer $a_t$ and $x_t$ from $s_t$ and $y_{t'}$, we first construct the mapping from $s_t$ and $y_{t'}$ to $a_t$ as:

$$a_t = f_\phi(s_t, y_{t'}) \tag{5}$$

where $f_\phi(\cdot)$ represents the multiscale encoder network parametrized by a neural network with parameters $\phi$.

One obstacle in the design of the encoder network is accounting for the different timescales without using augmentation techniques such as zero-padding as commonly done (Lipton et al., 2016; Zhu et al., 2021) since they can yield suboptimal performance and distort the information during latent factor inference (Wells et al., 2013; Che et al., 2018; Luo et al., 2018). Thus, it is important to account for timescale differences and the possibility of missing samples when designing the multiscale encoder. Additionally, our goal is to perform multimodal fusion at each time-step while also allowing for real-time inference of factors. We address these problems with a multiscale encoder network design shown in Figure 2.

In our multiscale encoder (Figure 2), first, each modality ($s_t$ and $y_{t'}$) is processed by separate multilayer perception (MLP) networks with parameters $\phi_s$ and $\phi_y$ to obtain modality-specific embedding factors, $a_t^s \in \mathbb{R}^{n_a}$ and $a_t^y \in \mathbb{R}^{n_a}$, respectively. Then, we construct modality-specific LDMs for each modality, whose observations are their corresponding embedding factors:

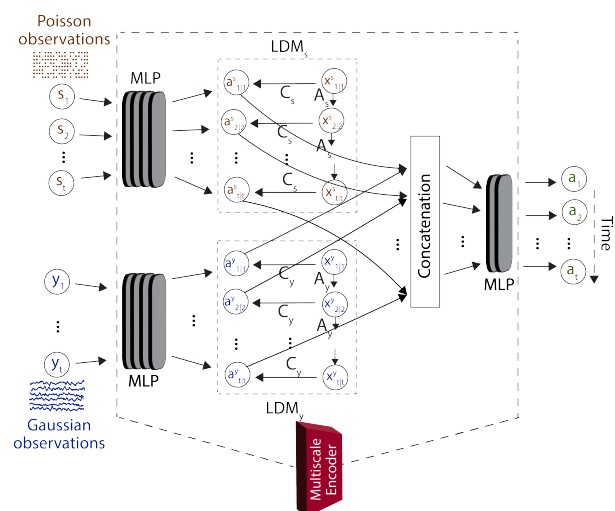

$$x_{t+1}^s = A_s x_t^s + w_t^s$$
$$a_t^s = C_s x_t^s + r_t^s \qquad (6)$$
$$x_{t+1}^y = A_y x_t^y + w_t^y$$
$$a_t^y = C_y x_t^y + r_t^y \qquad (7)$$

where $x_t^s, x_t^y \in \mathbb{R}^{n_x}$ are modality-specific latent factors, $A_s, A_y \in \mathbb{R}^{n_a \times n_a}$ are the state transition matrices, $C_s, C_y \in \mathbb{R}^{n_a \times n_a}$ are the emission matrices, $w_t^s, w_t^y \in \mathbb{R}^{n_a}$ are the zero-mean dynamics noises with covariances $W_s, W_y \in \mathbb{R}^{n_a \times n_a}$, and $r_t^s$ and $r_t^y$ are the zero-mean Gaussian observation noises with covariances $R_s, R_y \in$

Figure 2: MRINE multiscale encoder design. Modality-specific LDMs learn within-modality dynamics and account for timescale differences or missing samples via Kalman filtering. Then, filtered modality-specific embedding factors ($a_{t|t}^s$ and $a_{t|t}^y$) are fused and processed by another fusion network to obtain the multiscale embedding factors $a_t$. As an example, when $y_2$ is missing, $a_{2|2}^y$ is predicted only by the dynamics of modality-specific LDM in Eq. 7.

$\mathbb{R}^{n_a \times n_a}$ for modalities $s_t$ and $y_{t'}$, respectively. We denote the modality-specific LDM parameters by $\psi_s = \{A_s, C_s, W_s, R_s\}$ and $\psi_y = \{A_y, C_y, W_y, R_y\}$ for $s_t$ and $y_{t'}$, respectively.

In our design, the modality-specific LDMs allow us to account for missing samples whether due to timescale differences or missed measurements by using the learned within-modality state dynamics to predict these samples forward in time, while maintaining the operation fully real-time/causal. Specifically, given the modality-specific LDMs in Eq. 6 and 7, we can obtain the modality-specific latent factors, $x_{t|t}^s$ and $x_{t|t}^y$ with Kalman filtering, which is real-time and constitutes the optimal minimum mean-squared error estimator for these models (Kalman, 1960). We use the subscript $i|j$ to denote the factors inferred at time $i$ given all observations up to time $j$. As such, subscripts $t|t$, $t|T$ and $t+k|t$ denote causal/real-time filtering, non-causal smoothing and $k$-step-ahead prediction, respectively. At this stage, if $t$ is an intermittent time-step such that $y_t$ is missing (i.e., $t \in \mathcal{T}$ and $t \notin \mathcal{T}'$), $x_{t|t}^y$ is obtained with forward prediction using the Kalman predictor as $x_{t|t}^y = A_y x_{t-1|t-1}^y$ (Åström, 1970), and similarly for $x_{t|t}^s$ if $s_t$ is missing. Having done this for each modality, we perform information fusion by concatenating the modality-specific embedding factors and passing them through a fusion network with parameters $\phi_m$ to obtain the initial representation for $a_t$, which later becomes the noisy observations of the multiscale LDM formed by Eq. 1 and 2 (also see Figure 1). We denote the learnable multiscale encoder network parameters by $\phi = \{\phi_s, \phi_y, \psi_s, \psi_y, \phi_m\}$.

The multiscale LDM now allows us to infer $x_{t|t}$ with real-time (causal) Kalman filtering, or infer $x_{t|T}$ with non-causal Kalman smoothing (Rauch et al., 1965). Similarly, $k$-step-ahead predicted multiscale

latent factors $\boldsymbol{x}_{t+k|t}$ can be obtained by forward propagating $\boldsymbol{x}_{t|t}$ $k$-times into the future with Eq. 1, i.e., $\boldsymbol{x}_{t+k|t} = \boldsymbol{A}^k \boldsymbol{x}_{t|t}$. We denote the parameters of the multiscale LDM by $\psi_m = \{\boldsymbol{A}, \boldsymbol{C}, \boldsymbol{W}, \boldsymbol{R}\}$. We can now obtain the filtered, smoothed, and $k$-step-ahead predicted parameters of the likelihood functions in Eq. 3 and 4 by first using Eq. 2 to compute the corresponding multiscale embedding factors – i.e. $\boldsymbol{a}_{i|j} = \boldsymbol{C} \boldsymbol{x}_{i|j}$, where $i|j$ is $t|t$, $t|T$ and $t+k|t$, respectively – and then forward passing these factors through each modality's decoder network parametrized by $\theta_s$ or $\theta_y$ (Figure 1).

## 2.3 LEARNING THE MODEL PARAMETERS

$k$**-step ahead prediction** To learn the MRINE model parameters and encourage learning the dynamics, as part of the loss, we employ the multi-horizon $k$-step-ahead prediction loss defined as:

$$\mathcal{L}_k = -\sum_{k \in \mathcal{K}} \Big( \sum_{\substack{t \in \mathcal{T} \\ t \geq k}} \tau \log \left( p_{\theta_s}(\boldsymbol{s}_t \mid \boldsymbol{a}_{t|t-k}) \right) + \sum_{\substack{t' \in \mathcal{T}' \\ t' \geq k}} \log \left( p_{\theta_y}(\boldsymbol{y}_{t'} \mid \boldsymbol{a}_{t'|t'-k}) \right) \Big) \tag{8}$$

where $\mathcal{T}$ and $\mathcal{T}'$ denote the time-steps when $\boldsymbol{s}_t$ and $\boldsymbol{y}_{t'}$ are observed, respectively. $\tau$ is the scaling parameter as the log-likelihood values of different modalities are of different scales (see Appendix B.2), and $\mathcal{K}$ is the set of future prediction horizons. We note that $k$-step-ahead prediction is performed by computing $k$-step-ahead predicted multiscale latent factors, $\boldsymbol{x}_{t+k|t}$, rather than modality-specific ones.

**Smoothed reconstruction** In addition to the $k$-step-ahead prediction, we also optimize the reconstruction from smoothed multiscale factors:

$$\mathcal{L}_{smooth} = -\Big( \sum_{t \in \mathcal{T}} \tau \log \left( p_{\theta_s}(\boldsymbol{s}_t \mid \boldsymbol{a}_{t|T}) \right) + \sum_{t' \in \mathcal{T}'} \log \left( p_{\theta_y}(\boldsymbol{y}_{t'} \mid \boldsymbol{a}_{t'|T}) \right) \Big) \tag{9}$$

where $T$ is the last time-step that any modality is observed.

**Smoothness regularization** To impose a smoothness prior on learned dynamics and to prevent the model from overfitting by learning trivial identity encoder/decoder transformations, in our loss, we also apply smoothness regularization on $p_{\theta_s}(\boldsymbol{s}_t \mid \boldsymbol{a}_{1:T})$, $p_{\theta_y}(\boldsymbol{y}_{t'} \mid \boldsymbol{a}_{1:T})$ and $p(\boldsymbol{x}_t \mid \boldsymbol{a}_{1:T})$ by minimizing the KL-divergence between the distributions in consecutive time-steps as introduced in Li et al. (2021) for Gaussian-distributed modalities. Here, we extend this technique also to Poisson-distributed modalities. Let $\mathcal{L}_{sm}$ be the smoothness regularization penalty, defined as:

$$\begin{aligned} \mathcal{L}_{sm} = & \gamma_s \underbrace{\sum_{i=1}^{|\mathcal{T}|-1} \sum_{j=1}^{n_s} d\Big( p_{\theta_s}(s_{\mathcal{T}_i}^j \mid \boldsymbol{a}_{\mathcal{T}_i|T}), p_{\theta_s}(s_{\mathcal{T}_{i+1}}^j \mid \boldsymbol{a}_{\mathcal{T}_{i+1}|T}) \Big)}_{\text{Smoothness on } \boldsymbol{s}_t} \\ & + \gamma_y \underbrace{\sum_{i=1}^{|\mathcal{T}'|-1} \sum_{j=1}^{n_y} d\Big( p_{\theta_y}(y_{\mathcal{T}_i'}^j \mid \boldsymbol{a}_{\mathcal{T}_i'|T}), p_{\theta_y}(y_{\mathcal{T}_{i+1}'}^j \mid \boldsymbol{a}_{\mathcal{T}_{i+1}'|T}) \Big)}_{\text{Smoothness on } \boldsymbol{y}_{t'}} \\ & + \gamma_x \underbrace{\sum_{t=1}^{T} \sum_{j=1}^{\lfloor \frac{n_x}{2} \rfloor} d\Big( p(x_t^j \mid \boldsymbol{a}_{t|T}), p(x_{t+1}^j \mid \boldsymbol{a}_{t+1|T}) \Big)}_{\text{Smoothness on } \boldsymbol{x}_t} \end{aligned} \tag{10}$$

where $d(\cdot, \cdot)$ is the KL-divergence between given distributions, subscript $i$ denotes the $i^{\text{th}}$ element of the set, superscript $j$ is the $j^{\text{th}}$ component of the vector (e.g., $\boldsymbol{s}_{\mathcal{T}_i}$), and $p_{\theta_s}(\boldsymbol{s}_t \mid \boldsymbol{a}_{t|T})$ and $p_{\theta_y}(\boldsymbol{y}_{t'} \mid \boldsymbol{a}_{t'|T})$ are as in Eq. 3 and 4, respectively. Here $\gamma_s$, $\gamma_y$ and $\gamma_x$ are the scaling hyperparameters. The smoothness penalties on $\boldsymbol{s}_t$ and $\boldsymbol{y}_{t'}$ are computed over the time-steps that they are observed. The penalty on $\boldsymbol{x}_t$ is obtained over all time-steps as $\boldsymbol{x}_t$ is inferred for all time-steps. After extracting $\boldsymbol{a}_{1:T}$ with multiscale encoder, $p(\boldsymbol{x}_t \mid \boldsymbol{a}_{1:T})$ can be obtained with the Kalman smoother, which provides the posterior distribution for the multiscale LDM (Kalman, 1960; Dehghannasiri et al., 2018):

$$p(\boldsymbol{x}_t \mid \boldsymbol{a}_{1:T}) = \mathcal{N}(\boldsymbol{x}_t; \boldsymbol{x}_{t|T}, \Sigma_{t|T}) \tag{11}$$

where $\boldsymbol{x}_{t|T}$ and $\Sigma_{t|T}$ are the smoothed multiscale latent factors and their error covariances, respectively. To allow the model to learn both fast and slow dynamics, we put the smoothness regularization on $\boldsymbol{x}_t$ on half of its dimensions.

To assess the impact of incorporating smoothness regularization terms in Eq. 10 and smoothed reconstruction in Eq. 9, we performed an ablation study (see Appendix E.2) which demonstrates that each term contributes to the improved performance. Further, in the ablation study on the effect of multiscale modeling (see Appendix E.4), we show that MRINE's multiscale encoder design is another important contributing factor to improved performance, compared to the case where missing samples are imputed by zeros and removed from the training objectives above.

Finally, we form the loss as the sum of the above elements and regularization terms, and minimize it via mini-batch gradient descent using the Adam optimizer (Kingma & Ba, 2015) to learn the model parameters $\{\phi, \psi, \theta_s, \theta_y\}$:

$$\mathcal{L}_{MRINE} = \mathcal{L}_k + \mathcal{L}_{smooth} + \mathcal{L}_{sm} + \gamma_r L_2(\theta_s, \theta_y, \phi_s, \phi_y) \tag{12}$$

where $L_2(\cdot)$ is the $L_2$ regularization penalty on the MLP weights and $\gamma_r$ is the scaling hyperparameter.

Moreover, we employ a dropout technique termed *time-dropout* during training to increase the robustness of MRINE to missing samples even further. See Appendix A for more information, Appendix E.1 for an ablation study on the effect of *time-dropout*, and Appendix B for training details and hyperparameters.

## 3 RESULTS

### 3.1 STOCHASTIC LORENZ ATTRACTOR SIMULATIONS

We first validated that MRINE can successfully aggregate information across multiple modalities by performing simulations with the stochastic Lorenz attractor dynamics defined in Eq. 18. To do so, we generated Poisson and Gaussian observations with 5, 10 and 20 dimensions as described in Appendix C.2. We generated 4 systems with different random seeds and performed 5-fold cross-validation for each system. Then, we trained MRINE as well as single-scale networks with either only Gaussian observations or only Poisson observations (see Appendix B.1). To assess MRINE's ability to aggregate multimodal information, we compared its latent reconstruc-

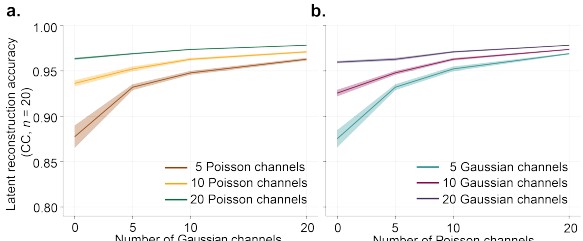

Figure 3: Latent reconstruction accuracies for the stochastic Lorenz attractor simulations. **a.** Accuracies when 5, 10, or 20 Poisson channels were the primary modality to which Gaussian channels were gradually added. Solid lines show the mean and shaded areas represent the standard error of the mean (SEM). **b.** Similar to **a**, but with the Gaussian channels being the primary modality.

tion accuracies with those of single-scale networks. For each model, these accuracies were obtained by computing the average correlation coefficient (CC) between the true and reconstructed latent factors (see Appendix C.3 for details). We refer to each observation dimension as a channel for simplicity.

We first considered the Poisson modality as the primary modality and fused with it 5, 10, and 20 Gaussian channels as the secondary modality. As shown in Figure 3a, for all different numbers of Poisson channels, gradually fusing Gaussian channels significantly improved the latent reconstruction accuracies ($p < 10^{-5}$, $n = 20$, one-sided Wilcoxon signed-rank test). As expected, these improvements were higher in the low information regime where fewer primary Poisson channels were available (5 and 10) compared to the high information regime (20 Poisson channels). Likewise, when the Gaussian modality was the primary modality, latent reconstruction accuracies again showed consistent improvement across all regimes as Poisson channels were fused (Figure 3b, $p < 10^{-5}$, $n = 20$, one-sided Wilcoxon signed-rank test).

## 3.2 MRINE FUSED MULTISCALE INFORMATION IN BEHAVIOR DECODING FOR THE NHP GRID REACHING DATASET

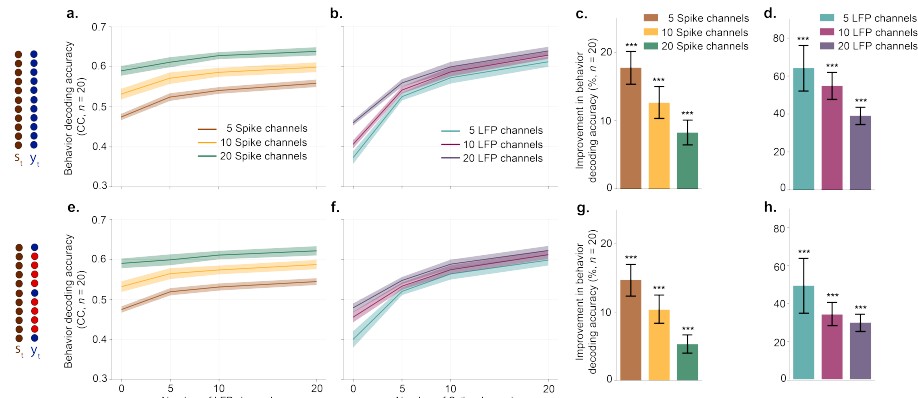

Figure 4: Behavior decoding accuracies for the NHP grid reaching dataset when spike and LFP channels had the same (*top row*) and different (*bottom row*) timescales. Shaded areas and error bars represent SEM. **a.** Accuracies when 5, 10, or 20 spike channels were the primary modality and an increasing number of LFP channels were fused. **b.** Similar to **a** when LFP channels were the primary modality. **c.** Percentage improvements in decoding accuracy when 20 LFP channels were added to 5, 10, and 20 spike channels. Asterisks indicate significance of comparison (***: $p < 0.0005$, one-sided Wilcoxon signed-rank test). **d.** Similar to **c**, when 20 spike channels were added to 5, 10, and 20 LFP channels. **e–h**. Same as **a–d** but when spike and LFP channels had different timescales.

To test MRINE's information aggregation capabilities in a real dataset, we used a publicly available NHP dataset (Makin et al., 2018; O'Doherty et al., 2020). In this dataset, discrete spiking activity and LFP signals were recorded while the subject was performing sequential 2D reaches on a grid to random targets appearing in a virtual-reality environment (Makin et al., 2018; O'Doherty et al., 2020) (details in Appendix D.1). We considered the 2D cursor velocity in the x and y directions as our target behavior variables to decode from inferred latent factors (see Appendix D.3 for details). We trained single-scale models with 5, 10, and 20 channels of spike and LFP signals, and MRINE models for every combination of these multimodal channel sets. In our analyses, we used 4 experimental sessions recorded on different days and performed 5-fold cross-validation for each session.

First, we tested MRINE's behavior decoding performance when both spike and LFP modalities had the same timescale, i.e., were observed every 10 ms (we abbreviate these LFPs as 10 ms LFPs). When spiking signals were taken as the primary modality, fusing them with increasing numbers of LFP channels steadily improved the behavior decoding accuracy for all different numbers of primary spike channels (Figure 4a, $p < 0.007$, $n = 20$, one-sided Wilcoxon signed-rank test). Similar to simulation results, improvements in behavior decoding accuracy were higher in the low-information regimes, i.e., for 5 and 10 primary spike channels. For instance, adding 20 LFP channels to 5, 10, and 20 spike channels improved behavior decoding accuracy by $17.7\%$, $12.6\%$, and $8.3\%$, respectively (Figure 4c). Similarly, when LFP signals were considered as the primary modality, behavior decoding accuracies again improved significantly when spike channels were fused (Figure 4b, $p < 10^{-4}$, $n = 20$, one-sided Wilcoxon signed-rank test). As illustrated in Figure 4d, the improvements in this scenario were higher compared to when spiking activity was the primary modality, i.e., adding 20 spike channels to 5, 10 and 20 LFP channels improved behavior decoding accuracy by $64.2\%$, $54.9\%$, and $39.1\%$, respectively (Figure 4c), indicating that spiking activity encoded more information than LFP signals about the target behavior in this dataset. Overall, the bidirectional improvements here suggest that spike and LFP modalities may encode non-redundant information about behavior variables that can be fused nonlinearly with MRINE.

We next tested MRINE when modalities had different timescales, i.e., spikes were observed every 10 ms and LFPs every 50 ms (abbreviated as 50 ms LFPs) (Hsieh et al., 2018; Ahmadipour et al., 2023). As expected, fusing spikes with 50 ms LFPs resulted in smaller improvements in behavior decoding compared to fusion with 10 ms LFPs. Nevertheless, as shown in Figure 4e,f, MRINE again improved behavior decoding accuracies significantly when LFP channels were added to spike channels ($p < 0.05$, $n = 20$, one-sided Wilcoxon signed-rank test) and spike channels were added to

LFP channels ($p < 10^{-4}$, $n = 20$, one-sided Wilcoxon signed-rank test). For instance, adding 20 channels of 50 ms LFP improved behavior decoding accuracy by $14.7\%$, $10.4\%$, and $5.3\%$ when fused with 5, 10, and 20 spike channels (Figure 4g), respectively. Compared to the previous case, percentage improvements had a maximum decrease of 3 percentage points, even when MRINE was trained with 5 times fewer LFP samples. These findings show that MRINE can aggregate multimodal information even when modalities have different timescales; MRINE utilizes the information in the undersampled (slower timescale) modality to infer faster timescale latent factors that are more predictive of the downstream task (here behavior decoding). We also tested MRINE on another publicly available NHP dataset (Flint et al., 2012), for which, we modeled discrete spiking activity (at every 10 ms) and LFP power signals (at every 10 or 50 ms) as our Poisson and Gaussian modalities. As shown in Fig. 7, MRINE again improved behavior decoding accuracy when spiking activity and LFP power signals are fused together, across all information regimes and both in the same and different timescale scenarios.

### 3.3 MRINE's information aggregation was robust to missing samples

Next, we studied the robustness of MRINE inference to missing samples. Here, in addition to having different timescales, we used various sample-dropping probabilities to drop spike or LFP samples in the NHP grid reaching dataset. For models that were trained with 20 channels of both spikes and 50 ms LFP, we fixed the dropping probability for LFP samples as 0.2 while varying that of spike samples (Figure 5a), and vice versa (Figure 5b) during inference. Behavior decoding accuracies of MRINE remained robust, decreasing by only $4.3\%$ and $17\%$ when $40\%$ and $80\%$ of spike samples were missing, respectively, in addition to $20\%$ of LFP samples missing (Figure 5a). Further, behavior decoding accuracies were more robust to missing LFP samples (Figure

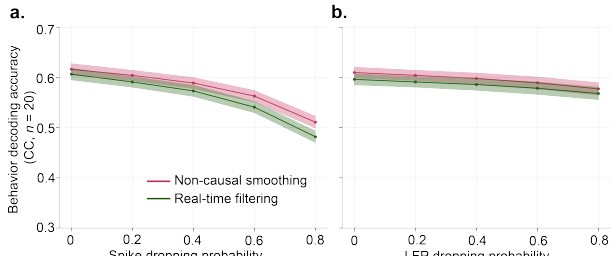

Figure 5: Behavior decoding accuracies in the NHP grid reaching dataset when spike and LFP channels had both missing samples and different timescales for MRINE. **a.** Accuracies when the sample dropping probability of LFPs was fixed at 0.2 while that of spikes was varied as shown on the x-axis. Lines represent mean and shaded areas represent SEM. **b.** Similar to **a** when sample dropping probability of spikes was fixed at 0.2 while that of LFPs was varied.

5b), again indicating the dominance of spiking activity in behavior decoding for this dataset. Finally, both causal filtering and non-causal smoothing were robust to missing samples.

In addition to behavior decoding, we also evaluated MRINE's information aggregation capabilities in cross-modal imputed predictions of spike and LFP signals. Similar to the previous case, in addition to having different timescales, we dropped one modality with various sample dropping probabilities where the other modality was fully observed. We found that MRINE outperforms single-scale models in neural cross-modal prediction even when up to $60\%$ of spike samples (Figure 8a) and $40\%$ of LFP samples were dropped (Figure 8b) where single-scale model predictions were obtained with fully available observations. See Appendix D.6 for more details.

### 3.4 MRINE improved behavior decoding compared with prior multimodal modeling methods

We compared MRINE with prior multimodal methods, namely the recent MSID in Ahmadipour et al. (2023), mmPLRNN in Kramer et al. (2022), MMGPVAE in Gondur et al. (2023) and MVAE in Wu & Goodman (2018). Further, we extended the unimodal LFADS model (Pandarinath et al., 2018) to the multimodal setting (denoted by mmLFADS) by modeling spikes and LFPs with separate observation models (similar to Eq. 3 and 4). We also trained LFADS models on multimodal data with the same likelihood by treating the multimodal data as a single modality, the results of which are denoted by LFADS below. Please see the ablation study on the effect of different observation models in Appendix E.3.

We trained MRINE, MSID, and MVAE with 5, 10, or 20 channels of spikes and 50 ms LFPs as they allow for different timescales but we used the same timescale (10 ms) for both spike and LFP signals to train mmPLRNN, LFADS, mmLFADS and MMGPVAE as they do not account for different timescales. Further, mmPLRNN, LFADS, mmLFADS and MMGPVAE decodings were performed non-causally unlike MRINE - real time and MSID since mmPLRNN, LFADS, mmLFADS and MMGPVAE do not support real-time recursive inference. For all numbers of primary channels (i.e., all information regimes), MRINE significantly outperformed both MSID, mmPLRNN, MVAE, LFADS, mmLFADS and MMGPVAE in behavior decoding (Table 1, $p < 0.06$, $n = 20$, one-sided Wilcoxon signed-rank test).

| Method | 5 Spike 5 LFP | 10 Spike 10 LFP | 20 Spike 20 LFP |
|---|---|---|---|
| MVAE | 0.326 ± 0.011 | 0.386 ± 0.009 | 0.425 ± 0.009 |
| MSID | 0.380 ± 0.021 | 0.440 ± 0.015 | 0.519 ± 0.012 |
| mmPLRNN | 0.455 ± 0.012 | 0.478 ± 0.011 | 0.533 ± 0.012 |
| LFADS | 0.467 ± 0.017 | 0.495 ± 0.015 | 0.548 ± 0.011 |
| mmLFADS | 0.468 ± 0.016 | 0.507 ± 0.015 | 0.547 ± 0.011 |
| MMGPVAE | 0.424 ± 0.012 | 0.511 ± 0.014 | 0.579 ± 0.009 |
| MRINE | 0.487 ± 0.007 | 0.555 ± 0.011 | 0.611 ± 0.012 |
| MRINE - noncausal | **0.519** ± **0.009** | **0.573** ± **0.011** | **0.621** ± **0.011** |

Table 1: Behavior decoding accuracies for the NHP grid reaching dataset with 5, 10, and 20 spike and LFP channels for MRINE, MSID , mmPLRNN, MVAE, LFADS, mmLFADS and MMGPVAE. The best-performing method is in bold, the second best-performing method is underlined, ± represents SEM.

We also trained mmPLRNN, LFADS, mmLFADS, and MMGPVAE with different timescale signals where missing LFP samples are imputed by their global mean, i.e., zero-imputation due to z-scoring. As shown in Table 10, the performance improvements of MRINE over baseline methods grew even further, indicating the importance of multiscale modeling. In addition, we compared MRINE's behavior decoding performance with missing samples with the dynamical baseline methods in Fig. 6 and show that MRINE outperforms all baseline methods across all missing sample regimes. Please also see Appendix D.7 for trial-averaged latent factor visualizations, Appendix B for more details on MSID, mmPLRNN, LFADS, mmLFADS, MMGPVAE and MVAE benchmarks. Also, see Table 6 for baseline comparisons for the NHP center-out reaching dataset.

## 4 RELATED WORK

**Single-Scale Models of Neural Activity**  Numerous dynamical models of neural activity have been developed. Some of these models are in linear, generalized linear, or switching linear form (Macke et al., 2011; Buesing et al., 2012; Cunningham & Yu, 2014; Kao et al., 2015; Koh et al., 2023; Linderman et al., 2017). LDMs are widely used in real-time applications because they provide real-time and recursive inference algorithms. However, LDMs cannot capture the potential nonlinearities underlying neural activity. For this reason, there has been an increased interest in deep learning architectures including recurrent neural network (RNN) based methods with nonlinear temporal dynamics (Pandarinath et al., 2018; She & Wu, 2020), autoencoder-based architectures that utilize the Markovian-property of linear dynamics to learn a smoothing distribution (Archer et al., 2015; Gao et al., 2016; Durstewitz, 2017), transformer encoder based models optimized with masked training (Ye & Pandarinath, 2021; Le & Shlizerman, 2022) and neural ordinary differential equations (Kim et al., 2021). These models have shown great promise in improving behavior decoding compared to the linear models (Pandarinath et al., 2018). A recent work Abbaspourazad et al. (2023) has also developed an autoencoder-based nonlinear framework with linear dynamics that supports real-time inference by utilizing Kalman filtering similarly with linear approaches (Kalman, 1960). Despite the LDM-based approaches can handle missing samples as well as some other nonlinear methods such as (Ramchandran et al., 2021), all these methods are designed for a single modality of neural activity and do not address multiscale modeling.

**Multimodal Information Fusion**  Outside neuroscience applications, fusing multiple modalities has been researched across many areas including natural language processing (NLP) and computer vision. It has been shown that integrating visual and/or acoustic signals with text can improve the performance of downstream classification tasks in NLP, e.g., sentiment analysis (Hazarika et al., 2020; Tsai et al., 2019a; Han et al., 2021; Poria et al., 2017; Zadeh et al., 2017). In computer vision, many studies focused on variational autoencoders (VAE), and approximated the joint posterior distribution by factorization over modality-specific posteriors (Wu & Goodman, 2018; Shi et al.,

2019; Sutter et al., 2021; Tsai et al., 2019b; Lee & Pavlovic, 2021) to handle missing modalities, or by concatenating the modality-specific representations (Suzuki et al., 2016; Vedantam et al., 2018). Instead of learning the common embedding space by factorization or concatenation, some studies have also employed cross-modality generation (Joy et al., 2021; Pandey & Dukkipati, 2017). Even though some of these methods can handle time-series modalities with different timescales, they need to do so using separate networks to noncausally encode each modality into a single vector. However, real-time applications require aggregating information at each time step, potentially from modalities with different timescales, in a causal/real-time manner to perform continuous decoding of targets.

**Multimodal Models in Neuroscience** A line of work for multimodal modeling in neuroscience aims to model single-scale neural activity and behavior as multimodal signals (Sani et al., 2021; Hurwitz et al., 2021; Schneider et al., 2023). However, their latent factor inference is performed by processing single-scale neural activity, similar to the single-scale models discussed above. Several approaches have been proposed for multiscale modeling of neural dynamics. However, these methods are either simply linear/generalized-linear or they are designed for offline reconstruction without enabling real-time inference. Specifically, some approaches proposed multimodal modeling frameworks (Rezaei et al., 2023; Coleman et al., 2011) that utilize linear temporal dynamics and learn the model parameters via expectation-maximization (EM). However, their latent factor inference is not designed to operate on multimodal signals with different timescales. To address this, a multiscale linear dynamical modeling framework for continuous LFP and discrete spiking activity has been introduced in Abbaspourazad et al. (2021), where model parameters are also learned with EM. With a similar linear formulation of multiscale dynamics, recent work in Ahmadipour et al. (2023) proposed a more computationally efficient learning framework compared to Abbaspourazad et al. (2021) by using subspace identification and showed that the method also performs more accurately than the EM-based approach. However, both approaches are of linear form and cannot characterize nonlinearities.

Recent studies Brenner et al. (2022); Kramer et al. (2022); Zhou & Wei (2020); Gondur et al. (2023) have developed a nonlinear multimodal framework that can process multimodal neural signals for latent factor inference, but their inference network is non-causal and thus does not enable real-time inference of latent factors. Further, their formulation assumes the same timescale for the different modalities and does not consider missing samples. Thus, in such situations, they would need to rely on indirect approaches such as augmentations with zero-padding, which can be suboptimal by changing the value of missing samples (Che et al., 2018; Wells et al., 2013; Zhu et al., 2021; Abbaspourazad et al., 2023). Instead, here we develop a nonlinear dynamical modeling framework for multimodal time-series data that supports real-time and efficient recursive inference, and handles both timescale differences and missing samples by directly leveraging the learned dynamical model to predict these missing samples.

## 5 DISCUSSION

In this study, we presented MRINE that can nonlinearly and in real-time aggregate information across multiple time-series modalities, even with different timescales or with missing samples. To achieve this, we proposed a novel multiscale encoder design that first extracts modality-specific representations while accounting for their timescale differences and missing samples, and then performs nonlinear fusion to aggregate multimodal information. We combined this encoder with a multiscale LDM backbone to achieve real-time multiscale fusion. Through stochastic Lorenz attractor simulations and real NHP datasets, we show MRINE's ability to causally fuse information across modalities even with different timescales or with missing samples. We show that MRINE outperforms recent linear and nonlinear multimodal methods, and these comparisons in addition to ablation studies on the effect of loss terms, and using different observation models show the importance of MRINE's multiscale nonlinear fusion and training objective outlined in Section 2.3. Further, as shown in Appendices E.4 and D.5, multiscale modeling is crucial for modalities with different timescales, as performances of current approaches significantly degrade with data imputation. A current limitation of MRINE is the assumption of time-invariant multiscale dynamics, which may not hold in non-stationary cases. In such cases, MRINE models may need to be intermittently retrained across days/sessions. Extending MRINE to track temporal variability such as with switching dynamics or adaptive approaches is an important future direction. Overall, we showed that MRINE enables real-time multiscale decoding and cross-modality prediction that can handle both timescale differences and missing samples, while also providing competitive performance. Thus MRINE can be especially important for future real-time systems such as brain-computer interfaces.

## 6 REPRODUCIBILITY STATEMENT

We provide all details on training MRINE and baseline models in Appendix B, including hyperparameters used for training. We also provide the implementation of MRINE, and configuration files to run MRINE models at this anonymous GitHub repository. We also provide the simulation details in Appendix C and references to the public dataset used in this study with the preprocessing steps in Appendix D.1. Details of our analyses are explained throughout the Appendices and the main text.

## 7 ETHICS STATEMENT

We develop a multiscale dynamical model of different time-series modalities to improve downstream real-time target decoding performance. The work can have many societal implications by improving the accuracy and robustness of neurotechnologies, e.g., brain-machine interfaces, for the treatment of brain disorders in millions of patients. We do not anticipate any negative societal impacts of our work.

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

# APPENDICES

## A   TIME-DROPOUT

To improve the robustness of the MRINE against missing samples, we developed a regularization technique denoted as "*time-dropout*". Before training MRINE, based on the availability of the observations denoted by $\mathcal{T}$ and $\mathcal{T}'$ (see Section 2), we define mask vectors $\boldsymbol{m}^s, \boldsymbol{m}^y \in \mathbb{R}^T$ for $\boldsymbol{s}_t$ and $\boldsymbol{y}_t$ respectively:

$$m_t^y = \begin{cases} 1, & \text{if } \boldsymbol{y}_t \text{ is observed at } t \text{ (i.e., if } t \in \mathcal{T}' \\ 0, & \text{otherwise} \end{cases} \tag{13}$$

$$m_t^s = \begin{cases} 1, & \text{if } \boldsymbol{s}_t \text{ is observed at } t \text{ (i.e., if } t \in \mathcal{T} \\ 0, & \text{otherwise} \end{cases} \tag{14}$$

where subscript $t$ is the $t^{\text{th}}$ component of the vector. The general assumption is that $\mathcal{T}' \subseteq \mathcal{T}$, such that $\boldsymbol{s}_t$ is available for all time-steps where $\boldsymbol{y}_t$ is observed – this choice is motivated by the faster timescale of spikes compared with field potentials. However, this scenario may not always hold as recording devices can have independent failures leading to dropped samples at any time. To mimic the partially missing scenario where either modality can be missing, as well as the fully missing scenarios where both can be missing, we randomly replaced (dropped) elements of $\boldsymbol{m}^s$ and $\boldsymbol{m}^y$ by 0 at every training step, with the same dropout probabilities $\rho_t$ for both modalities. Masked time-steps (time-steps with 0 mask value) were not used either during the latent factor inference described in Section 2.2 or in the computation of loss terms in Eq. 8 and 9 so that the inference and model learning were not distorted by missing samples. Instead, note that our inference procedure uses the learned model of dynamics to account for missing samples during both inference and learning. We note that we used the original masks (before applying *time-dropout*) while computing $\mathcal{L}_{sm}$ in Eq. 10, as we wanted to obtain smooth representations for all available observations. Note that *time-dropout* differs from masked training that is commonly used for training transformer-based networks. Masked training aims to predict masked samples from existing samples unlike our training objective (see Section 2.3 for details). Here, the goal of *time-dropout* is to increase the robustness to missing samples by artificially introducing partially and fully missing samples during model training, rather than being the training objective itself. See the ablation study on the effect of *time-dropout* in Appendix E.1 which shows that MRINE models trained with *time-dropout* had more robust behavior decoding performance and the effect of *time-dropout* was more prominent with more missing samples.

In addition to the *time-dropout*, we also applied regular dropout (Srivastava et al., 2014) in the encoder's input and output layers with probability $\rho_d$.

## B   TRAINING DETAILS

### B.1   TRAINING SINGLE-SCALE NETWORKS

For the case of single-scale networks, we use a special case of the architecture in Fig. 1 by replacing the multiscale encoder with an MLP encoder, and by just using a single-scale LDM with one modality's decoder network. In particular, for field potentials, we use a Gaussian decoder , and for the spikes we use a Poisson decoder to obtain the corresponding likelihood distribution parameters. Unlike MRINE's modality-specific LDMs used in the multiscale encoder, the single-scale networks have an MLP. Also, instead of the multiscale LDM of MRINE, this MLP is then followed by a single-scale LDM to perform inference both in real-time and non-causally. During learning of the single-scale networks, we dropped the terms related to the discarded modality from loss functions in Eq. 8, 9 and 10.

### B.2   SETTING THE LIKELIHOOD SCALING PARAMETER $\tau$

The log-likelihood values of modalities with different likelihood distributions may be of different scales, e.g., Poisson log-likelihood has a smaller scale than Gaussian log-likelihood in our case (and this can change arbitrarily by simply multiplying the Gaussian modality with any constant). To prevent the model from putting more weight on one modality vs. the other due to a higher

log-likelihood scale while learning the dynamics, we scaled the log-likelihood of the modality with a smaller scale by a parameter $\tau$. To set this parameter, we first computed the time averages of each modality over $\mathcal{T}$ and $\mathcal{T}'$ for $\boldsymbol{s}_t$ and $\boldsymbol{y}_{t'}$, respectively, as:

$$\boldsymbol{\lambda} = \frac{1}{|\mathcal{T}|} \sum_{t \in \mathcal{T}} \boldsymbol{s}_t \tag{15}$$

$$\boldsymbol{\mu} = \frac{1}{|\mathcal{T}'|} \sum_{t' \in \mathcal{T}'} \boldsymbol{y}_{t'} \tag{16}$$

where $\boldsymbol{\lambda} \in \mathbb{R}^{n_s}$ and $\boldsymbol{\mu} \in \mathbb{R}^{n_y}$. Then, we computed the corresponding log-likelihoods of $\boldsymbol{s}_t$ and $\boldsymbol{y}_{t'}$ by using these time averages as the means of their likelihood distributions (assuming unit variance for Gaussian distribution). Finally, we set $\tau$ as the ratio between the higher and smaller scale log-likelihoods:

$$\tau = \frac{\frac{1}{|\mathcal{T}'|} \sum_{t' \in \mathcal{T}'} \log(\mathcal{N}(\boldsymbol{y}_{t'}; \boldsymbol{\mu}, \mathbf{1}))}{\frac{1}{|\mathcal{T}|} \sum_{t \in \mathcal{T}} \log(\text{Poisson}(\boldsymbol{s}_t; \boldsymbol{\lambda}))} \tag{17}$$

where $\mathbf{1} \in \mathbb{R}^{n_y}$ is the 1's vector denoting the unit variances for the Gaussian likelihood. The above allows us to balance the contribution of both modalities in our loss function during learning.

## B.3 HYPERPARAMETERS

Tables 2, 3, 4 and 5 provide the hyperparameters and network architectures used for training single-scale networks and MRINE on stochastic Lorenz simulations and analyses of the real NHP datasets.

For all models, we used a cyclical learning rate scheduler (Smith, 2015) starting with a minimum learning rate of 0.001, and reaching the maximum learning rate of 0.01 in 10 epochs. The maximum learning rate is exponentially decreased by a scale of 0.99. Across all experiments, batch size was set to 32, MLP weights were initialized by Xavier-normal initialization (Glorot & Bengio, 2010), and tanh function was used as the activation function of hidden layers. All models were trained on CPU servers (AMD Epyc 7513 and 7542, 2.90 GHz with 32 cores) with parallelization.

| Models | Hyperparameters | | | | | | | | | | | | | | | |
|---|---|---|---|---|---|---|---|---|---|---|---|---|---|---|---|---|
| | $\phi_s$ | $\phi_y$ | $\phi_m$ | $\theta_s$ | $\theta_y$ | $n_a$ | $n_x$ | $\mathcal{K}$ | $\rho_t$ | $\rho_d$ | GC | $\gamma_s$ | $\gamma_y$ | $\gamma_x$ | $\gamma_r$ | TE |
| SS-Poisson | 3,128 | - | - | 3,128 | - | 32 | 32 | 1,2,3,4 | 0.3 | 0.4 | 0.1 | 100 | - | 30 | 0.0001 | 200 |
| SS-Gaussian | - | 3,128 | - | - | 3,128 | 32 | 32 | 1,2,3,4 | 0.3 | 0.4 | 0.1 | - | 50 | 30 | 0.0001 | 200 |
| MRINE | 3,128 | 3,128 | 1,128 | 3,128 | 3,128 | 32 | 32 | 1,2,3,4 | 0.3 | 0.4 | 0.1 | 250 | 10 | 30 | 0.001 | 200 |

Table 2: Hyperparameters used for the stochastic Lorenz attractor simulations. SS denotes single-scale network. We represent the architecture's various MLP encoders and decoders with their parameter notations, and for each, provide the number of hidden layers and hidden units in order, separated by commas. Specifically. $\phi_s$ and $\phi_y$ – i.e., MLP blocks through which $\boldsymbol{s}_t$ and $\boldsymbol{y}_{t'}$ are passed in Figure 2 – represent the modality-specific encoder networks for Poisson and Gaussian modalities, respectively. $\phi_m$ is the fusion network (the last MLP block in Figure 2). Modality-specific decoder networks are $\theta_s$ and $\theta_y$ for Poisson and Gaussian modalities, respectively. $n_a$ and $n_x$ represent dimensions of $\boldsymbol{a}_t$ and $\boldsymbol{x}_t$, respectively. $\mathcal{K}$ is the set of future prediction horizons in Eq. 8. $\rho_t$ is the time dropout probability on the mask vectors, and $\rho_d$ denotes the dropout probability applied in the input and output layers of the encoder network (see Appendix A). GC represents the global gradient clipping norm on learnable parameters. $\gamma_s$, $\gamma_y$, and $\gamma_x$ are scaling parameters for smoothness regularization penalty in Eq. 10. $\gamma_r$ is the $L_2$ penalty on MLP weights of the encoder and decoder networks. TE denotes the number of training epochs.

## B.4 MSID TRAINING

Multiscale subspace identification (MSID) is a recently proposed linear multiscale dynamical model of neural activity that assumes linear dynamics. MSID can handle different timescales and allow for real-time inference of latent factors (Ahmadipour et al., 2023). We compared MRINE with MSID and showed that compared with the linear approach of MSID, the nonlinear information

| Models | Hyperparameters | | | | | | | | | | | | | | | |
|---|---|---|---|---|---|---|---|---|---|---|---|---|---|---|---|---|
| | $\phi_s$ | $\phi_y$ | $\phi_m$ | $\theta_s$ | $\theta_y$ | $n_a$ | $n_x$ | $\mathcal{K}$ | $\rho_t$ | $\rho_d$ | GC | $\gamma_s$ | $\gamma_y$ | $\gamma_x$ | $\gamma_r$ | TE |
| SS-Poisson | 3,128 | - | - | 3,128 | - | 64 | 64 | 1,2,3,4 | 0.3 | 0.1 | 0.1 | 100 | - | 30 | 0.0001 | 500 |
| SS-Gaussian | - | 3,128 | - | - | 3,128 | 64 | 64 | 1,2,3,4 | 0.3 | 0.1 | 0.1 | - | 10 | 30 | 0.0001 | 500 |
| MRINE | 3,128 | 3,128 | 1,128 | 3,128 | 3,128 | 64 | 64 | 1,2,3,4 | 0.3 | 0.1 | 0.1 | 250 | 10 | 30 | 0.001 | 500 |

Table 3: Hyperparameters used for the NHP grid reaching dataset analysis with same timescales for both modalities. Hyperparameter definitions are the same as in Table 2.

| Models | Hyperparameters | | | | | | | | | | | | | | | |
|---|---|---|---|---|---|---|---|---|---|---|---|---|---|---|---|---|
| | $\phi_s$ | $\phi_y$ | $\phi_m$ | $\theta_s$ | $\theta_y$ | $n_a$ | $n_x$ | $\mathcal{K}$ | $\rho_t$ | $\rho_d$ | GC | $\gamma_s$ | $\gamma_y$ | $\gamma_x$ | $\gamma_r$ | TE |
| SS-Poisson | 3,128 | - | - | 3,128 | - | 64 | 64 | 1,2,3,4 | 0.3 | 0.1 | 0.1 | 100 | - | 30 | 0.0001 | 500 |
| SS-Gaussian | - | 3,128 | - | - | 3,128 | 64 | 64 | 1,2,3,4 | 0.3 | 0.1 | 0.1 | - | 5 | 30 | 0.0001 | 500 |
| MRINE | 3,128 | 3,128 | 1,128 | 3,128 | 3,128 | 64 | 64 | 1,2,3,4 | 0.3 | 0.1 | 0.1 | 250 | 5 | 30 | 0.001 | 500 |

Table 4: Hyperparameters used for the NHP grid reaching dataset analysis with different timescales for the different modalities. Hyperparameter definitions are the same as in Table 2.

| Models | Hyperparameters | | | | | | | | | | | | | | | |
|---|---|---|---|---|---|---|---|---|---|---|---|---|---|---|---|---|
| | $\phi_s$ | $\phi_y$ | $\phi_m$ | $\theta_s$ | $\theta_y$ | $n_a$ | $n_x$ | $\mathcal{K}$ | $\rho_t$ | $\rho_d$ | GC | $\gamma_s$ | $\gamma_y$ | $\gamma_x$ | $\gamma_r$ | TE |
| SS-Poisson | 3,128 | - | - | 3,128 | - | 64 | 64 | 1,2,3,4 | 0.3 | 0.1 | 0.1 | 30 | - | 30 | 0.0001 | 200 |
| SS-Gaussian | - | 3,128 | - | - | 3,128 | 64 | 64 | 1,2,3,4 | 0.3 | 0.1 | 0.1 | - | 5 | 30 | 0.0001 | 200 |
| MRINE | 3,128 | 3,128 | 1,128 | 3,128 | 3,128 | 64 | 64 | 1,2,3,4 | 0.3 | 0.1 | 0.1 | 50 | 5 | 30 | 0.001 | 200 |

Table 5: Hyperparameters used for the NHP center-out reaching dataset analysis with same and different timescales for the different modalities. Hyperparameter definitions are the same as in Table 2.

aggregation enabled by MRINE can improve downstream behavior decoding while still allowing for real-time recursive inference and for handling different timescales. To train MSID, we used the implementation provided by the authors and set the horizon hyperparameters with the values provided in their manuscript, i.e., $h_y = h_z = 10$. For the comparisons reported in Table 1, as recommended by the developers in their manuscript (Ahmadipour et al., 2023), we fitted MSID models with various latent dimensionalities consisting of [8, 16, 32, 64] and picked the one with the best behavior decoding accuracy found with inner-cross validation done on the training data.

### B.5 MMPLRNN TRAINING

Multimodal piecewise-linear recurrent neural networks (mmPLRNN) is a recent multimodal framework that assumes piecewise-linear latent dynamics coupled with modality-specific observation models (Kramer et al., 2022). As discussed in Section 4, mmPLRNN has shown great promise in reconstructing the underlying dynamical system but its inference network operates non-causally in time and assumes the same timescale between modalities, unlike MRINE. Therefore, for the comparisons provided in Table 1, the behavior decoding with mmPLRNN was performed non-causally and with neural modalities of the same timescale (10 ms) unlike MRINE and MSID. To train mmPLRNN, we used the variational inference training code provided by the authors in their manuscript. However, the default implementation of mmPLRNN only supports Gaussian and categorical distributed modalities. Thus, we implemented the Poisson observation model by following Appendix C of Kramer et al. (2022). Further, we replaced the default linear decoder networks with nonlinear MLP networks for a fair comparison and better performance.

Finally, we trained all mmPLRNN models for 100 epochs. We also performed hyperparameter searches for latent state dimension, learning rate, and number of neurons in encoder/decoder layers over grids of [16, 32, 64], [1e-4, 5e-4, 1e-3, 1e-2] and [32, 64, 128], respectively.

As shown in Table 1, MRINE outperformed mmPLRNN in behavior decoding for the NHP grid reaching dataset across all information regimes, even though MRINE was trained on neural modalities of different timescales. We believe that the future-step-ahead prediction training objective along with new smoothness regularization terms and smoothed reconstruction are important elements contributing to such performance gap (see Appendix E.2) whereas mmPLRNN is trained on optimizing evidence lower bound (ELBO), whose optimization can be challenging due to KL-divergence term (Dosovitskiy & Brox, 2016; Bowman et al., 2016; Razavi et al., 2019; Shao et al., 2020).

### B.6   (MM)LFADS TRAINING

LFADS is a sequential autoencoder-based model of unimodal neural activity proposed by Pandarinath et al. (2018). The results reported for LFADS were obtained by training LFADS models on concatenated multimodal data which corresponds to treating multimodal data as a single modality. Further, we extended LFADS to support multimodal neural activity with different observation models (denoted by mmLFADS), i.e. Poisson and Gaussian observation models for spikes and LFP, respectively. To achieve that, we replaced LFADS' unimodal decoder network that maps factors to observation model parameters with multimodal decoder networks similar to Eq. 3 and 4. We used the authors' codebase[2] to train LFADS models and implement the multimodal extension, i.e., mmLFADS. We used the hyperparameters in the second row of Supplementary Table 1 in Pandarinath et al. (2018) with a factor dimension of 64 and used the default learning rate scheduler and early stopping criterion used in the codebase. As shown in Appendix E.2 and E.4, we believe that MRINE's training objectives and multiscale encoder design are contributing factors to its improved performance over (mm)LFADS.

### B.7   MVAE TRAINING

Multimodal variational autoencoder (MVAE) is a variational autoencoder-based architecture proposed in Wu & Goodman (2018) that can account for partially paired multimodal datasets by a mixture of experts posterior distribution factorization. However, the notion of partial observations in our work and MVAE are different. In MVAE, partial observations refer to having partially missing data tuples in each element of the batch, which would translate to having completely missing LFP or spike signals for a given trial/segment of multimodal neural activity. However, as we detailed in Section 2, we are interested in modeling multimodal neural activities with different sampling rates, i.e., partially missing time-steps rather than missing either of the signals completely. Further, as discussed in Section 4, the latent factor inference in MVAE is designed to encode each modality to a single factor, whereas MRINE is designed to infer latent factors for each time-step so that behavior decoding can be performed at each time-step. To account for all these differences, we trained MVAE models without a dynamic backbone by treating each time-step as a different data point that allowed us to train MVAE models with partially missing time-steps as done for MRINE. As shown in Table 1, MVAE showed the lowest performance among all methods, which could be caused by lacking a dynamical backbone, unlike other methods.

### B.8   MMGPVAE TRAINING

Multimodal Gaussian process variational autoencoder (MMGPVAE) is another recent multimodal framework that utilizes Gaussian process to model latent distribution underlying multimodal observations (Gondur et al., 2023). Distinct from other approaches discussed in this work, MMGPVAE inference network extracts the frequency content of the latent factors followed by conversion to time domain representations, rather than direct estimation on the time domain. This approach allows MMGPVAE to prune high-frequency content in the latent factors that help in obtaining smooth representations. Similar to mmPLRNN, MMGVAE does not allow training on modalities with different timescales, thus, for the comparisons provided in Table 1, the behavior decoding with MMGPVAE was also performed non-causally and with neural modalities of the same timescale (10 ms). To train MMGPVAE, we used the authors' official implementation provided in their manuscript. To provide a fair comparison, we trained MMGPVAE models with 64-dimensional latent factors for each modality, where 32 dimensions of 64-dimensional latent factors were shared across modalities. All MMGPVAE models are trained for 100 epochs as behavior decoding performances reached their peak performance in around 100 epochs, then started degrading due to overfitting. The default encoder/decoder architecture for Gaussian modality in the MMGPVAE codebase was implemented for an image dataset, which resulted in poor performance in our dataset. Therefore, for the encoder/decoder architectures, we used the same architectures as MRINE. Further, we scaled the likelihood of Poisson modality with the same $\tau$ value as done for MRINE (see Section B.2). We also performed a hyperparameter search for the learning rate in a grid of [1e-3, 5e-4, 1e-4] since default values resulted in poor convergence. Despite MMGPVAE being the best competitor of MRINE as shown in Table 1 and Table 6, we believe

---

[2]https://lfads.github.io/lfads-run-manager/

that MRINE's training objective is an important factor contributing to its improved performance over MMGPVAE.

## C STOCHASTIC LORENZ ATTRACTOR SIMULATIONS

### C.1 DYNAMICAL SYSTEM

The following set of dynamical equations defines the stochastic Lorenz attractor system:

$$
\begin{aligned}
dx_1 &= \sigma(x_2 - x_1)dt + q_1 \\
dx_2 &= (\rho x_1 - x_1 x_3 - x_2)dt + q_2 \\
dx_3 &= (x_1 x_2 - \beta x_3)dt + q_3
\end{aligned}
\tag{18}
$$

where $x_1$, $x_2$, and $x_3$ are the latent factors of the Lorenz attractor dynamics, $dt$ denotes the discretization time-step of the continuous system and $d$ is the change of variables in $dt$ time. We used $\sigma = 10$, $\rho = 28$, $\beta = \frac{8}{3}$ and $dt = 0.006$ as in Pandarinath et al. (2018). $q_1$, $q_2$, and $q_3$ are zero-mean Gaussian dynamic noises with variances of 0.01. We generated 750 trials each containing 200 time-steps, and the initial condition of each trial was obtained by running the system for 500 burn-in steps starting from a random point. Then, we normalized trajectories to have zero mean and a maximum value of 1 across time for each latent dimension.

### C.2 GENERATING HIGH DIMENSIONAL OBSERVATIONS

To generate the Gaussian-distributed modalities, we multiplied the normalized latent factors by a random matrix $\boldsymbol{C}_y \in \mathbb{R}^{n_y \times 3}$ and added zero mean Gaussian noise with variance of 5 to generate noisy observations.

To generate the Poisson-distributed modalities, we first generated firing rates by multiplying the normalized trajectories by another random matrix $\boldsymbol{C}_s \in \mathbb{R}^{n_s \times 3}$ and added a log baseline firing rate of 5 spikes/sec with bin-size of 5 ms, followed by exponentiation. We then generated spiking activity by sampling from the Poisson process whose mean is the simulated firing rates.

### C.3 COMPUTING THE LATENT RECONSTRUCTION ACCURACY

For MRINE and each single-scale network trained only on Poisson or Gaussian modalities, we obtained the smoothed single-scale or multiscale latent factors $\boldsymbol{x}_{t|T}$, $t \in \{1, 2, \ldots, T\}$ for each trial in the training and test sets. To quantify how well the inferred latent factors can reconstruct the true latent factors, we fitted a linear regression model from the inferred latent factors of the training set to the corresponding true latent factors. Using the same linear regression model, we reconstructed the true latent factors from the inferred latent factors of the test set. Then, we computed the Pearson correlation coefficient (CC) between the true and reconstructed latent factors for each trial and latent dimension. The reported values are averaged over trials and latent dimensions.

## D REAL DATASET ANALYSIS

### D.1 NONHUMAN PRIMATE (NHP) GRID REACHING DATASET

In this publicly available dataset (Makin et al., 2018; O'Doherty et al., 2020), a macaque monkey performed a 2D target-reaching task by controlling a cursor in a 2D virtual environment. All experiments were performed in accordance with the US National Research Council's Guide for the Care and Use of Laboratory Animals and were approved by the UCSF Institutional Animal Care and Use Committee. Monkey I was trained to perform continuous reaches to circular targets with a 5 mm visual radius randomly appearing on an 8-by-8 square or an 8-by-17 rectangular grid. The cursor was controlled by the monkey's fingertips, and the targets were acquired if the cursor stayed within a 7.5 mm-by-7.5 mm target acceptance zone for 450 ms. Even though there was no inter-trial interval between sequential reaches, there existed a 200 ms lockout interval after a target acquisition during which no target could be acquired. After the lockout interval, a new target was randomly drawn from the set of possible targets with replacement. Fingertip position was recorded with a six-axis

electromagnetic position sensor (Polhemus Liberty, Colchester, VT) at 250 Hz and non-causally low-pass filtered to reject the sensor noise (4th order Butterworth, with 10 Hz cut-off frequency). The cursor position was computed by a linear transformation of the fingertip position, and we computed 2D cursor velocity using discrete differentiation of the 2D cursor position in the x and y directions. In our analysis, we used the 2D cursor velocity as the behavior variable to decode.

One 96-channel silicon microelectrode array (Blackrock Microsystems) was chronically implanted into the subject's right hemisphere primary motor cortex. Each array consisted of 96 electrodes, spaced at 400 $\mu$m and covering a 4mm-by-4mm area. We used multi-unit spiking activity obtained at a 10 ms timescale, and LFP signals were extracted from the raw neural signals by low-pass filtering with 300 Hz cut-off frequency, and downsampling to either 100 Hz (10 ms timescale) or 20 Hz (50 ms timescale). In our study, we picked the top spiking and LFP channels based on their individual behavior prediction accuracies and considered a maximum of 20 channels for each modality. As this dataset consists of continuous recordings without a clear trial structure, we created 1-second non-overlapping segments from continuous recordings to form trials so that we could utilize mini-batch gradient descent during model learning.

## D.2 NHP CENTER-OUT REACHING DATASET

In this publicly available dataset (Flint et al., 2012), a macaque monkey performed a 2D center-out reaching task while grasping a two-link manipulandum. All experiments were performed with approval from the Institutional Animal Care and Use Committee of Northwestern University. Monkey C was trained to perform reaches from a center position to 2-cm square outer targets in an 8-target environment, where outer targets were spaced at 45-degree intervals around a 10-cm radius circle. Each trial of the task started with the illumination of the center target where the monkey had to hold the manipulandum for a random hold time of 0.5-0.6 seconds. After, the center target disappeared and an outer target was randomly selected from the pool of possible 8 targets, which signaled the monkey to start the reach. To obtain the reward, the monkey had to reach the outer target within 1.5 seconds and hold the manipulandum at the outer target for a random time of 0.2-0.4 seconds. Then, the monkey returned back to the center target position and the next trial started. In our analysis, we used 2D manipulandum velocity as the behavior variable to decode.

One 96-channel silicon microelectrode array (Blackrock Microsystems) was chronically implanted into the subject's proximal arm area of primary motor (M1) and premotor (PMd) cortices contralateral to the arm used to perform the task. We used multi-unit spiking activity obtained at a 10 ms timescale. LFP signals in the original dataset were extracted from the raw neural signals by band-pass filtering between 0.5 and 500 Hz and sampled at 2 kHz. From these LFP signals, we computed LFP power signals with a window of size 256 ms (moved at 10 ms resolution) over 5 bands (0-4, 7-20, 70-115, 130-200 and 200-300 Hz), resulting in LFP power signals at 100 Hz. For the different timescale analyses, we downsampled LFP power signals to 20 Hz (50 ms timescale). The rest of the dataset generation details are the same as the previous dataset.

## D.3 BEHAVIOR DECODING

In our analysis, we took 2D cursor velocity in the x and y directions as the behavior variables for downstream decoding. After we inferred smoothed (or filtered) multiscale (or single-scale) latent factors $x_t$ from MRINE or single-scale networks for both training and test sets, we fitted a linear regression model from inferred latent factors of the training set to the corresponding behavior variables. Then, we used the same linear regression model to decode the behavior variables from the inferred latent factors in the test set. We quantified the behavior decoding accuracy by computing the CC between the true and reconstructed behavior variables across time and averaging over behavior dimensions. Unless otherwise stated, all decodings were performed from smoothed latent factors.

When MRINE was trained with spiking activity and LFP signals with timescales of 10 ms and 50 ms, respectively, behavior decoding was performed at the 10 ms timescale for comparisons between MRINE and single-scale networks trained with spike channels. To provide a fair comparison between MRINE and single-scale networks trained with 50 ms LFP signals, inferred latent factors of

MRINE were downsampled to 50 ms from 10 ms, and behavior was decoded at every 50 ms in these comparisons with LFP.

## D.4 BEHAVIOR DECODING WITH MISSING SAMPLES

In this analysis, we first trained MRINE models with 20 spike and 20 LFP channels with different timescales (whose behavior decoding accuracies are shown in Figure 4e,f when there were no missing samples). To test the robustness of MRINE to missing samples, we randomly dropped samples in time during inference with fixed sample dropping probabilities for both modalities. Then, we inferred latent factors at all time-steps using only the available observations after sample dropping and performed behavior decoding as described above. Note that even though time-series observations were missing in time, behavior variables were available for all time-steps and were decoded at all time-steps using MRINE's inference method that accounts for missing observations with the learned local dynamics.

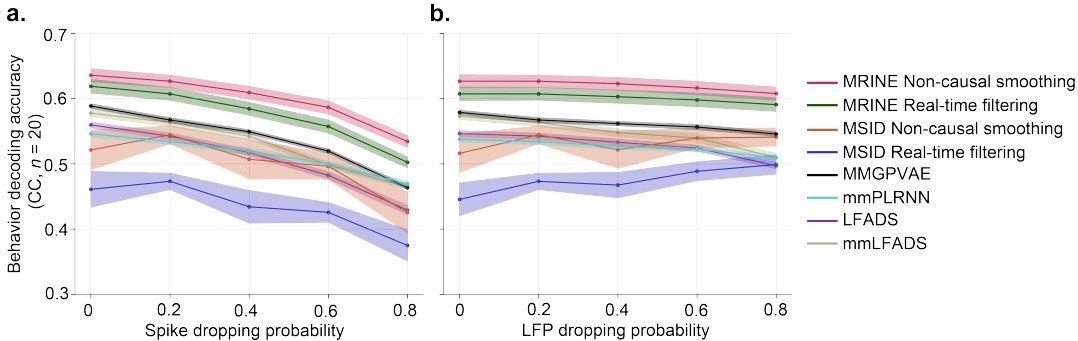

Figure 6: Behavior decoding accuracies all models for the NHP grid reaching dataset when spike and LFP channels had missing samples and the same timescale. **a.** Accuracies for models trained with 20 spike and 20 LFP channels. Sample dropping probability of LFPs was fixed at 0.2 while that of spikes was varied as shown on the x-axis. Lines represent mean and shaded areas represent SEM. **b.** Similar to **a** when sample dropping probability of spikes was fixed at 0.2 while that of LFPs was varied.

In addition to the different timescales scenario (Figure 5), we performed the same missing sample robustness analysis for all dynamical models (i.e., MRINE, MSID, mmPLRNN, LFADS, mmLFADS and MMGPVAE) trained with modalities of the same timescale. For all methods, we followed the same procedure described above after training models with same timescale signals. As shown in Figure 6, MRINE was again robust to missing samples for both modalities and outperformed all baseline dynamical models across all missing sample regimes. Compared to dropping 50 ms LFP samples, MRINE was more robust to missing LFP samples in this case where both LFPs and spikes had a 10 ms timescale, and thus LFPs were more abundant. Further, in general, the decoding was more robust to missing LFPs than missing spikes (compare panel b vs. a), which, consistent with earlier findings, again indicates the dominance of spiking activity in behavior decoding for this dataset.

## D.5 BEHAVIOR DECODING FOR THE NHP CENTER-OUT REACHING DATASET

We tested MRINE's information aggregation capabilities also in another real NHP dataset (Flint et al., 2012). In this dataset, discrete spiking activity and LFP power signals were recorded while the subject was performing center-out reaches to random targets (Flint et al., 2012) (details in Appendix D.2). We considered the 2D manipulandum velocity in the x and y directions as our target behavior variables to decode from inferred latent factors. We trained single-scale models with 5, 10, and 20 channels of spike and LFP power signals, and MRINE models for every combination of these multimodal channel sets. Note that for this dataset, each 5 power band signals are extracted for each LFP channel,

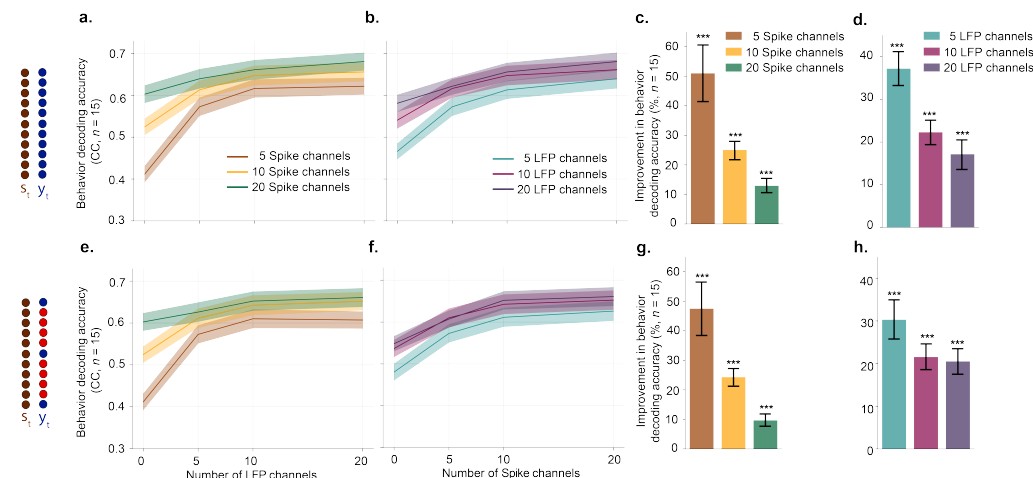

Figure 7: Behavior decoding accuracies for the NHP center-out reaching dataset when spike and LFP power signals had the same (*top row*) and different (*bottom row*) timescales. Figure convention is the same as Fig. 4.

thus, the input dimensionality for 5, 10, and 20 LFP channels are 25, 50, and 100. In our analyses, we used 3 experimental sessions recorded on different days and performed 5-fold cross-validation for each session.

Similar to our prior analysis on the grid reaching dataset, we tested MRINE's behavior decoding performance when both spike and LFP modalities had the same (10 ms for both) and different (10 ms for spikes and 50 ms for LFP power signals) timescales. In line with our previous results, MRINE was able to improve behavior decoding accuracy in a bidirectional manner, i.e., adding LFP power signals to spiking activity and vice versa, both in same and different timescale scenarios. However, unlike the grid reaching dataset, fusing LFP power signals with spiking activity resulted in higher improvements in behavior decoding accuracy, indicating that they encode more information about the target behavior compared to the first dataset.

| Method | 5 Spike 5 LFP | 10 Spike 10 LFP | 20 Spike 20 LFP |
|---|---|---|---|
| MSID | 0.444 ± 0.018 | 0.529 ± 0.021 | 0.571 ± 0.021 |
| mmPLRNN | 0.530 ± 0.022 | 0.556 ± 0.025 | 0.591 ± 0.027 |
| **LFADS** | 0.443 ± 0.019 | 0.450 ± 0.022 | 0.414 ± 0.022 |
| **mmLFADS** | 0.395 ± 0.026 | 0.443 ± 0.018 | 0.519 ± 0.025 |
| MMGPVAE | 0.558 ± 0.022 | 0.624 ± 0.022 | 0.670 ± 0.021 |
| MRINE | 0.550 ± 0.021 | 0.628 ± 0.022 | 0.664 ± 0.020 |
| MRINE - noncausal | **0.572** ± **0.022** | **0.647** ± **0.023** | **0.681** ± **0.021** |

Table 6: Behavior decoding accuracies for the NHP center-out reaching dataset with 5, 10, and 20 spike and LFP channels for MRINE, MSID, mmPLRNN, MVAE, LFADS, mmLFADS and MMGPVAE. The best-performing method is in bold, the second best-performing model is underlined, ± represents SEM.

Further, we compared MRINE's behavior decoding accuracy with dynamical baseline methods for center-out reaching dataset. For MRINE, we report behavior decoding accuracies both with real-time (causal) and noncausal latent factor inference procedures. Note that mmPLRNN, LFADS, mmLFADS,

and MMGPVAE perform noncausal latent factor inference as they do not support real-time inference.

As shown in Table 6, MRINE achieves the best behavior decoding performance among all methods with noncausal latent factor inference ($p < 0.006$, $n = 15$, one-sided Wilcoxon signed-rank test). When MRINE performs real-time (causal) latent factor inference, MMGPVAE outperforms MRINE across low (5 channels) and high (20 channels) information regimes where MRINE is slightly better in medium (10 channels) information regime, however, improvements are not statistically significant ($p = 0.08$ for 5 channels and MMGPVAE > MRINE, $p = 0.12$ for 10 channels and MRINE > MMGPVAE, and $p = 0.11$ for 20 channels and MMGPVAE > MRINE, $n = 15$, one-sided Wilcoxon-signed rank test).

Similar to the ablation study in Appendix E.4, we trained mmPLRNN and MMGPVAE models with different timescale signals where missing LFP power signals are imputed with their global mean, i.e., zero-imputation due to z-scoring, and removed from each model's training objective (from the likelihood calculations). In this scenario, performances of both mmPLRNN and MMGPVAE degraded significantly, whereas MRINE outperformed both methods significantly ($p < 10^{-6}$, $n = 15$, one-sided Wilcoxon-signed rank test), and it achieved comparable performance to that of in the same timescale scenario. Overall, these results again indicate the importance of multiscale modeling.

| Model | Behavior Decoding (CC) |
|---|---|
| MRINE | **0.661 ± 0.022** |
| mmPLRNN w/ Zero Imputation and Loss Masking | 0.538 ± 0.032 |
| MMGPVAE w/ Zero Imputation and Loss Masking | 0.530 ± 0.030 |

Table 7: Behavior decoding accuracies for the NHP center-out reaching dataset with 20 channels of 10 ms spike and 20 channels of 10 ms zero-imputed LFP signals for mmPLRNN and MMGPVAE where zero-imputed LFP time-steps are masked in the training objective. MRINE models are trained with different timescale signals as it supports multiscale training. The best-performing method is in bold, ± represents SEM.

### D.6    CROSS-MODAL IMPUTATION

To evaluate MRINE's information aggregation capabilities beyond behavior decoding, we computed cross-modal imputed one-step-ahead prediction accuracies of both modalities under various sample dropping probabilities in addition to having different timescales. To do that, the modality of interest was randomly dropped when the other modality was fully observed. Therefore, the one-step-ahead predictions of the missing modality were generated by leveraging the learned modality-specific and multiscale dynamics that allow for cross-modal imputations. Unlike MRINE, one-step-ahead prediction accuracies of single-scale networks were obtained with fully available observations.

For LFP signals modeled with Gaussian likelihood, we quantified the one-step-ahead prediction accuracy by computing the CC between the one-step-ahead predicted mean of the Gaussian likelihood distribution ($\boldsymbol{\mu}(\boldsymbol{a}_{t+1|t})$) and the true observations across time.

For spike signals modeled with Poisson likelihood, one-step-ahead prediction accuracy was quantified using the prediction power (PP) measure, which is defined as PP= 2AUC−1 where AUC denotes the area under the curve of the receiver operating characteristic (ROC) curve (Macke et al., 2011; Abbaspourazad et al., 2021). We constructed the ROC by using the one-step-ahead predicted firing rates, i.e., $\boldsymbol{\lambda}(\boldsymbol{a}_{t+1|t})$, as the classification scores to determine whether a time-step contained a spike or not (Truccolo et al., 2010). Both metrics were averaged over observation dimensions.

As shown in Figure 8, MRINE outperformed single-scale networks in one-step-ahead prediction accuracy even when 60% of spike samples (panel a) and 40% of LFP samples were missing. This result suggests that MRINE's information aggregation capabilities are not only limited to behavior decoding, but can also enable cross-modal prediction under missing sample scenarios.

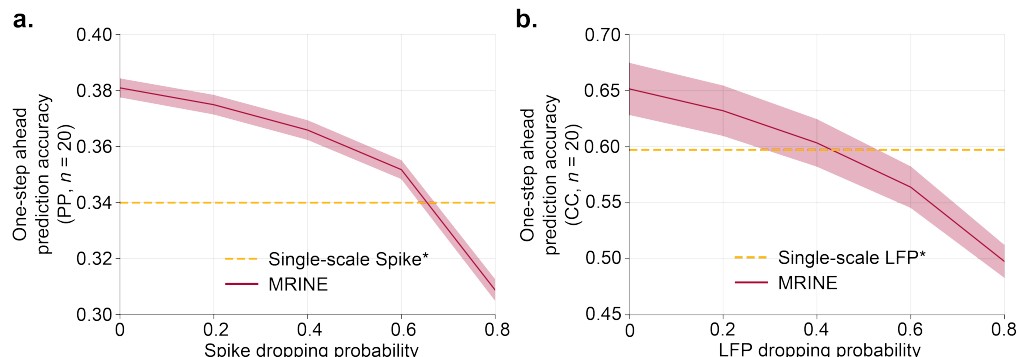

Figure 8: Cross-modal one-step-ahead prediction accuracies for the NHP grid reaching dataset when spike and LFP channels had different timescales. **a.** The one-step-ahead prediction accuracies of the spike channels. Sample dropping probability of spikes was varied as shown on the x-axis while LFP channels were fully available. The yellow dashed line represents the single-scale network's performance with fully available spike channels. Lines represent mean and shaded areas represent SEM. **b.** Similar to **a** when sample dropping probability of LFPs was varied while spike channels were fully available.

### D.7 VISUALIZATIONS OF LATENT DYNAMICS

To compute trial-averaged 3D PCA visualizations, for each algorithm, we first computed 3D PCA projections of latent factors, split them based on trial start and end indices, interpolated them to a fixed length (due to variable-length trials), and then computed trial averages of PCA projections for each of 8 different reach directions. As expected based on literature (Pandarinath et al., 2015; Kalidindi et al., 2021), all models recovered rotational neural population dynamics (see Fig. 9). Among all these algorithms, MRINE had the most clear rotations.

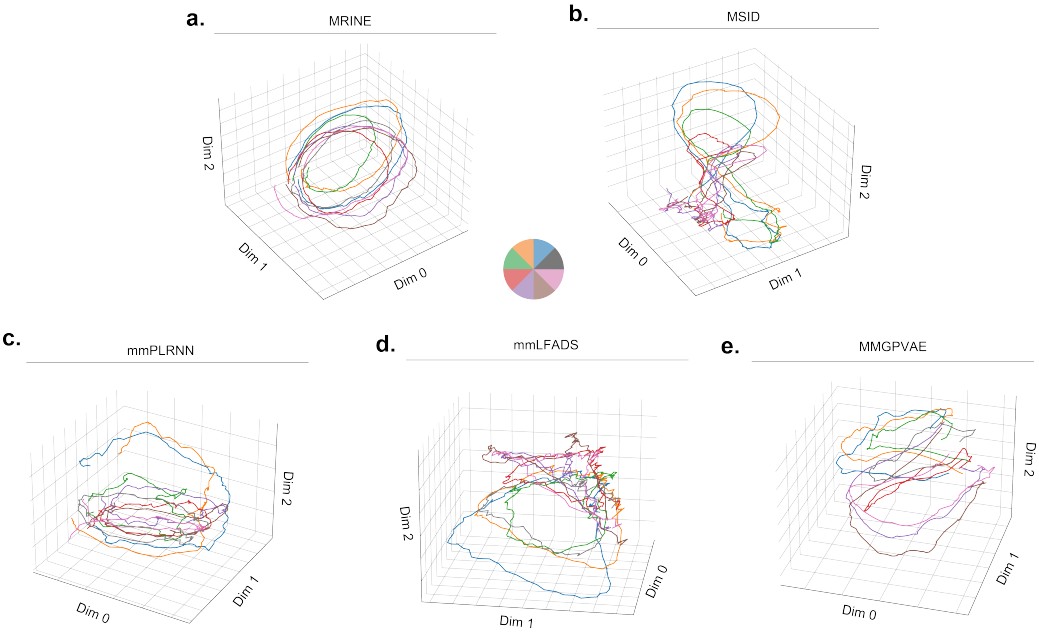

Figure 9: 3D PCA visualizations of trial-averaged latent factors inferred by **a)** MRINE, **b)** MSID, **c)** mmPLRNN, **d)** mmLFADS and **e)** MMGPVAE.

# E ABLATION STUDIES

## E.1 EFFECT OF TIME-DROPOUT

To test the effectiveness of *time-dropout*, we performed an ablation study with the same setting used to generate Fig. 5 (see Section 3.3) but we disabled *time-dropout* ($\rho_t = 0$). The remaining hyperparameters were as in Table 4. As shown in Fig. 10a, without *time-dropout*, the behavior decoding accuracies of MRINE decreased by 6.3% and 28.9% when 40% and 80% of spike samples were missing (in addition to 20% of LFP samples missing), whereas MRINE models trained with *time-dropout* experienced smaller performance drops of 4.3% and 17% in the same missing sample settings (see Figure 5). Similarly, MRINE models trained with *time-dropout* were more robust to missing LFP samples (Fig. 10b vs. Fig. 5b) but the performance drops were smaller due to spiking activity being the dominant modality for behavior decoding in this dataset. As expected, the effect of *time-dropout* was more prominent in the high sample dropping probability regimes (i.e., more missing samples).

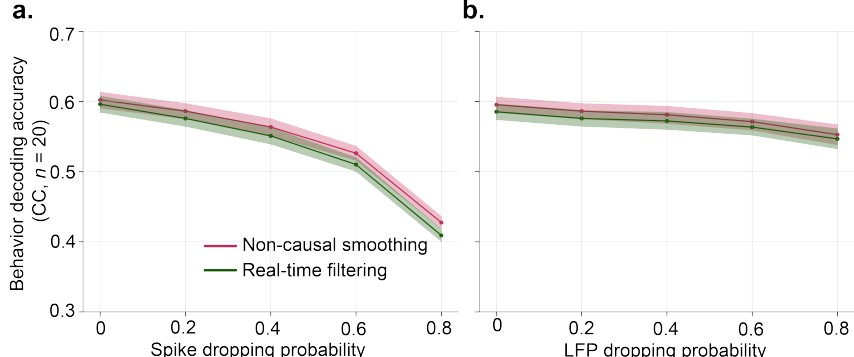

Figure 10: Behavior decoding accuracies in the NHP grid reaching dataset when *time-dropout* was disabled ($\rho_t = 0$), and spike and LFP channels had both missing samples and different timescales. The figure conventions are the same as in Fig. 5 .

## E.2 EFFECT OF LOSS TERMS IN THE BEHAVIOR DECODING PERFORMANCE

To gain intuition on the improved performance with MRINE, we performed an ablation study on the effect of smoothness regularization terms in Eq. 10 and smoothed reconstruction term in Eq. 9 on behavior decoding performance. To achieve that, we trained MRINE models with 20 channels of 10 ms spike and 20 channels of 10 ms LFP signals by removing Eq. 9 and individual terms in Eq. 10 from the training objective in Eq. 12. Then, we performed behavior decoding with these MRINE models as described in Section D.3.

As shown in Table 8, both smoothness regularization terms in Eq. 10 and smoothed reconstruction term in Eq. 9 are important factors contributing to improved behavior decoding performance as MRINE model trained without Eq. 9 and 10 (row 2) achieve worse performance than that of mmPLRNN and mmLFADS in Table 1. We observed that applying smoothness regularization on $x_t$ is an important contributing factor to improved performance (row 3 vs row 5) as well as smoothing reconstruction term (row 2 vs row 3). Even though smoothness regularizations on $s_t$ and $y_{t'}$ may seem marginal when comparing results in rows 3 and 4, they play a crucial role when combined with smoothness regularization on $x_t$ (comparing row 1 and row 5).

## E.3 EFFECT OF USING DIFFERENT OBSERVATION MODELS

While designing models of multimodal neural activity, a natural design choice is to treat modalities with different statistical properties as a single modality with the same observation model, e.g., modeling spikes with Gaussian likelihood as LFPs. Even though this modeling approach would not be effective when modalities are recorded with different timescales due to requiring imputation for missing time-steps, it would allow the training of unimodal models of neural activity on multimodal neural signals when recorded with the same timescale.

| Model | Behavior Decoding (CC) |
|---|---|
| MRINE | **0.621 ± 0.010** |
| MRINE w/o Eq. 9 and Eq. 10 | 0.524 ± 0.013 |
| MRINE w/o Eq. 10 | 0.565 ± 0.012 |
| MRINE w/o $\boldsymbol{x}_t$ in Eq. 10 | 0.566 ± 0.016 |
| MRINE w/o $\boldsymbol{s}_t$ and $\boldsymbol{y}_{t'}$ in Eq. 10 | 0.598 ± 0.012 |

Table 8: Behavior decoding accuracies for the NHP grid reaching dataset with 20 channels of 10 ms spike and 20 channels of 10 ms LFP signals for MRINE models trained without (w/o) loss terms denoted in the first column. The best-performing method is in bold, ± represents SEM.

To test the effect of treating each modality with their corresponding observation models, we trained MRINE models with 20 channels of spike and LFP signals sampled at 10 ms resolution where we modeled spikes with Gaussian or Poisson observation models, as done for LFADS and mmLFADS results in Table 1 (see Appendix B.6 for details).

| Model | Behavior Decoding (CC) |
|---|---|
| MRINE | **0.621 ± 0.010** |
| MRINE w/ Same Observation Model | 0.606 ± 0.009 |
| LFADS | 0.548 ± 0.011 |
| mmLFADS | 0.547 ± 0.011 |

Table 9: Behavior decoding accuracies for the NHP grid reaching dataset with 20 channels of 10 ms spike and 20 channels of 10 ms LFP signals for MRINE and LFADS models trained with same and different observation models. The best-performing method is in bold, ± represents SEM.

As shown in Table 9, MRINE models trained with different observation models, i.e., Poisson and Gaussian observation models for spikes and LFPs, respectively, outperforms MRINE models trained with Gaussian observation model. We observed that LFADS performance does not improve with mmLFADS when spiking activity is treated with a separate Poisson observation model, unlike MRINE. However, the performance gap between MRINE models trained with different and same observation models indicate that treating the modalities with appropriate observation models can indeed improve performance.

### E.4    EFFECT OF MULTISCALE ENCODER DESIGN

As discussed in Section 2.2, accounting for different sampling rates for neural signals is an important consideration for MRINE's encoder design shown in Fig. 2. To achieve that, we learn modality-specific LDMs in MRINE's encoder that can leverage within-modality state dynamics to account for missing samples whether due to timescale differences or missed measurements. Therefore, MRINE can perform inference without relying on augmentations to impute missing samples, such as zero-imputation as done in common practice (Lipton et al., 2016; Zhu et al., 2021) that can yield suboptimal performance (Che et al., 2018; Wells et al., 2013).

To investigate this, we upsampled 50 ms LFP signals to 10 ms with zero-imputation (note that zero-imputation translates to mean-imputation since LFP signals are z-scored before training) and trained MRINE, mmPLRNN and mmLFADS models with 20 channels of 10 ms spike and 10 ms zero-imputed LFP signals. For all models, we trained 2 versions where reconstructions/predictions of zero-imputed LFP time-steps are either included ($\mathcal{T}' = \mathcal{T}$) or masked ($\mathcal{T}' \subset \mathcal{T}$) in the training

objective. For both versions of MRINE, zero-imputed LFP time-steps were not treated as missing samples during latent factor inference (even if they were masked in the training objective for the second version).

| Model | Behavior Decoding (CC) |
|---|---|
| MRINE | $\mathbf{0.611 \pm 0.012}$ |
| MRINE w/ Zero Imputation | $0.523 \pm 0.013$ |
| MRINE w/ Zero Imputation and Loss Masking | $0.581 \pm 0.014$ |
| mmPLRNN w/ Zero Imputation | $0.498 \pm 0.009$ |
| mmPLRNN w/ Zero Imputation and Loss Masking | $0.539 \pm 0.011$ |
| mmLFADS w/ Zero Imputation | $0.280 \pm 0.022$ |
| mmLFADS w/ Zero Imputation and Loss Masking | $0.253 \pm 0.023$ |
| MMGPVAE w/ Zero Imputation | $0.393 \pm 0.023$ |
| MMGPVAE w/ Zero Imputation and Loss Masking | $0.518 \pm 0.011$ |

Table 10: Behavior decoding accuracies for the NHP grid reaching dataset with 20 channels of 10 ms spike and 20 channels of 10 ms zero-imputed LFP signals for MRINE, mmPLRNN, mmLFADS and MMGPVAE where zero-imputed LFP time-steps are either included or masked in the training objective. The best-performing method is in bold, $\pm$ represents SEM.

As shown in Table 10, the performance of MRINE models trained with zero-imputed LFP signals degraded compared to MRINE models trained with 50 ms LFP signals (row 1 vs rows 2 and 3). As expected, masking zero-imputed LFP time-steps in the loss function improved behavior decoding performance compared to the scenario where they are included in the loss function (row 2 vs row 3). However, removing zero-imputed LFP time-steps from the training objective still results in degraded performance for MRINE, showing the importance of multiscale encoder design (row 1 vs row 3).

Even though the performance of mmPLRNN improved significantly when zero-imputed LFP time-steps were masked in the training objective (row 4 vs row 5), allowing it to achieve performance comparable to that in Table 1, mmLFADS performance did not show a similar improvement. This is likely caused by mmLFADS' inference procedure that requires summarization of the input signal into an initial condition for the generator network. In contrast, mmPLRNN, MRINE and MMGPVAE do not require such a summarization process. Therefore, even if zero-imputed LFP time-steps are discarded in the training objective, they can still distort the inference procedure and distort and yield suboptimal performance. Unlike mmPLRNN, MMGPVAE performance experienced a higher decline in its performance compared to that of in Table 1, with zero-imputed LFP time-steps, potentially due to distorting the frequency content of the signal significantly. As expected, when zero-imputed LFP time-steps were not discarded in the training objective, MMGPVAE performance degraded even further. Overall, MRINE significantly outperformed all models when they were trained with zero-imputed different timescale signals, including its own variants ($p < 0.0003$, $n = 20$, one-sided Wilcoxon-signed rank test). These results show the importance of encoder design when modeling modalities with different timescales.

