# OpenReview forum: "Dynamical modeling for real-time inference of nonlinear latent factors in multiscale neural activity"
_ICLR.cc/2025/Conference — Submitted to ICLR 2025_

### Official Review · Reviewer_9Zgw · 2024-10-31

**Soundness:** 3
**Presentation:** 3
**Contribution:** 2
**Rating:** 5
**Confidence:** 3

**Summary:**

The paper, "MRINE: A Novel Framework for Real-Time Decoding of Target Variables from Multiscale Neural Time-Series Data," presents a new approach aimed at addressing multimodal neural data processing challenges. While the proposed framework could contribute to advancements in real-time decoding, several critical issues and areas for improvement weaken the paper's rigor, interpretability, and reproducibility.

**Strengths:**

The efforts to handle missing data with techniques like smoothness regularization and time-dropout are noteworthy additions to the model.

**Weaknesses:**

-(Lines 64-65): The claim that no existing models address multimodal neural data is inaccurate. Recent works, including those by Rezaei et al. [1] and Coleman et al. [2], have tackled similar challenges.
- (Lines 78-80): The assertion that other models do not address different timescales and missing samples in multimodal data is also incorrect. Models like those by García et al. [3] incorporate stochasticity in dynamical systems, relevant to noise handling in neural data.

-The paper lacks citations and comparisons to relevant models, particularly dynamical VAEs, which bear strong conceptual similarities to MRINE. This oversight diminishes the theoretical foundation of the paper and obscures recent advancements in variational autoencoders (VAEs) applied to dynamical systems.

-The equations in Section 2.1 contain inaccuracies, particularly in the notation used to describe multiscale dynamics. For instance, a single time index t is used, with no discussion on how different time horizons are managed, which undermines clarity regarding multiscale capabilities.

- (Equation 8): The absence of a derivation for the loss function weakens the rigor of the methodology section. Without a clear mathematical path to this objective function, readers are left without the means to evaluate or reproduce the proposed approach accurately.

-The paper lacks a comprehensive comparison with state-of-the-art models in time-series decoding, such as Mamba and S4. The selective comparison hinders the assessment of MRINE’s capabilities relative to these established baselines.

-Figure 1 does not effectively illustrate the multiscale time dynamics central to MRINE’s approach, diminishing its ability to convey the model’s core features.

-The bar plots throughout the paper lack statistical tests and error bars, raising questions about the reliability of reported performance differences. Including these would strengthen the study’s claims and clarify the significance of observed results.

-The theoretical foundation for MRINE’s multiscale capabilities is unclear, particularly in the equations defining these aspects. Some claims made in the introduction, especially those motivating the MRINE framework, appear overstated in light of prior work by Rezaei et al. [1], Coleman et al. [2], and García et al. [3].

-While these techniques are positive additions for handling missing data, they require further explanation, particularly regarding how they compare or improve upon existing methods for handling missing samples in neural data.


To strengthen the paper's contribution to the field, the following improvements are recommended:

Provide accurate citations and address the existence of related multimodal models that tackle similar challenges.
Include mathematical derivations for critical components, such as the loss function, to improve rigor and reproducibility.
Conduct comparative analyses with state-of-the-art models, particularly those in time-series decoding, to provide a balanced evaluation of MRINE’s capabilities.
Add statistical tests and error bars in visualizations to enhance the reliability of reported findings.
Clarify the use and advantages of techniques like smoothness regularization and time-dropout relative to existing methods.
References
[1] Rezaei, Mohammad R., et al. "Inferring cognitive state underlying conflict choices in verbal Stroop task using heterogeneous input discriminative-generative decoder model." Journal of Neural Engineering 20.5 (2023): 056016.

[2] Coleman, Todd P., et al. "A mixed-filter algorithm for dynamically tracking learning from multiple behavioral and neurophysiological measures." The dynamic brain: An exploration of neuronal variability and its functional significance (2011): 3-28.

[3] García, Constantino A., et al. "Stochastic embeddings of dynamical phenomena through variational autoencoders." Journal of Computational Physics 454 (2022): 110970.

**Questions:**

-Could you clarify the novelty of MRINE relative to existing multimodal neural data models? For instance, how does it differ from models like those presented by Rezaei et al. [1] and Coleman et al. [2] that also address multimodal data processing?

- Given that MRINE bears similarities to dynamical VAEs, could you discuss how MRINE’s approach compares to these methods, particularly regarding handling multiscale dynamics?

-To provide a fuller evaluation, would you consider adding comparisons with state-of-the-art time-series decoding models like Mamba and S4?

-The equations in Section 2.1 describe MRINE’s multiscale capabilities, but the use of a single time index t leaves the handling of distinct time horizons unclear. Could you provide additional details or adjustments to the notation to clarify how the model represents multiple time scales?

-The log-likelihood calculation in Figure 3.C is not clearly explained. Could you provide a detailed description of the calculation process and how it relates to model performance?

---

> ### Author Response · Authors · 2024-11-21
>
> > [W1] The claim that no existing models address multimodal neural data is inaccurate. Recent works, including those by Rezaei et al. [1] and Coleman et al. [2], have tackled similar challenges.
>
> We thank the reviewer for their comment. First, we should clarify an important point. We do not claim that no existing models address multimodal neural data. Indeed, we performed baseline comparisons to MSID (Ahmadipour et al., 2023) and mmPLRNN (Kramer et al., 2022) (and now MMGPVAE by Gondur et al., 2023) as baseline multimodal models, and **we acknowledged all these methods as multimodal models of neural data (see introduction and related work)**. Further, we refer to other multimodal models of neural data both in our introduction and related work. Rather, we claim that prior nonlinear multimodal models of neural activity are not developed **for modalities recorded with different timescales, which is a main capability enabled by MRINE**. Similar to all multimodal neural data models that we gave references to, the references that the reviewer provides (Rezaei et al., 2023, Coleman et al., 2011) also do not address this challenge of different time-scales (sometimes we refer to this as **multiscale** modeling).
>
> - As an important note, we believe that the ‘different temporal scale’ terminology between neural and behavior signals used in Rezaei et al., 2023 does not refer to having modalities recorded at different time-steps as there is no mention of different recording frequencies nor usage of different time indices in the work. Rather, they refer to different modalities having slower/faster dynamics as they say: “... behavioral signals … are usually very low dimensional and have slow temporal dynamics compared to neural signals …”. This is also achieved with MRINE through modality-specific dynamics, in addition to having different timescales.
>
> Finally, regarding both Rezaei et al., 2023 and Coleman et al., 2011, both works provide a generalized linear multimodal model of neural data and learn the model parameters with expectation-maximization (EM) (despite Rezaei et al., 2023 discusses potential nonlinearities, they say that they the discriminative process is linear, which is crucial for inferring nonlinear latent factors). However, as mentioned above, they were not developed for modalities with different timescales. In our work, we gave references to 2 recent works on generalized linear multimodal modeling, which are Abbaspourazad et al., 2021 and Ahmadipour et al., 2023, that can also model different timescale modalities. Similar to Rezaei et al., 2023 and Coleman et al., 2011, Abbaspourazad et al., 2021 learn the model parameters with EM, whereas Ahmadipour et al., 2023 formulate the model learning with subspace identification. As Ahmadipour et al., 2023 show improved performance over EM for generalized linear multimodal modeling (i.e. over Abbaspourazad et al., 2021), we provided comparisons to Ahmadipour et al., 2023 in our results, which is termed MSID throughout the text. We now added both Rezaei et al., 2023 and Coleman et al., 2011 to where we cite these two works on multimodal generalized modeling and discuss it there. We apologize for missing these originally.
>
> Furthermore, even though Rezaei et al., 2023 work with multiple neural modalities such as LFP and EEG, one difference of MRINE is modeling modalities with distinct stochastic profiles, e.g., discrete spiking activity with Poisson and continuous LFPs with Gaussian observation model, respectively, where Rezaei et al., 2023 propose using the same, e.g. Gaussian, observation model for LFP and EEG signals. This is potentially achieved through concatenating different neural signals as if they are a single modality since they are observed simultaneously with the same timescale. However, such an approach would not be possible in different timescale scenario without data imputations (which have suboptimal performance as we show in GR Tables 1 and 3).

---

> ### Author Response · Authors · 2024-11-21
>
> > [W2] The assertion that other models do not address different timescales and missing samples in multimodal data is also incorrect. ...
>
> We agree with the reviewer that Garcia et al., 2022 incorporates learnable temporal stochasticity in state dynamics, but as the reviewer seems to agree, Garcia et al., 2022 **is not designed for multimodal signals, let alone for multimodal signals of different timescales (i.e., multiscale modalities)**. We clarify that we do not claim that noise handling or stochasticity is the contribution of MRINE. Indeed, as we cite and discuss in our introduction and related work, many models of neural data incorporate temporal stochasticity in their state dynamics (e.g., all the linear or generalized linear dynamical models). Also, linear/generalized linear multimodal dynamical models can address missing samples through Kalman filtering, as we for example acknowledge for MSID. However, linear models usually perform worse than nonlinear/deep-learning models due to lacking nonlinearity. As such, the contribution of MRINE is to develop a **nonlinear model of multiscale neural data** that **1)** achieves real-time inference, **2)** addresses different time-scales, and **3)** handles missing samples. Prior deep-learning models of multimodal neural data do not achieve these capabilities.
>
> &nbsp;
>
> > [W3] The paper lacks citations and comparisons to relevant models, particularly dynamical VAEs, which bear strong conceptual similarities to MRINE ...
>
> It seems that there may have been a miscommunication in our text, and we thank the reviewer for bringing this to our attention. We clarify that we agree with the reviewer and **indeed provide references and comparisons to dynamical VAEs in our work**.
> - For instance, LFADS and mmLFADS are sequential/dynamical variational autoencoders.
> - mmPLRNN is another instance of dynamical VAE that utilizes piecewise linear dynamics and optimizes the ELBO objective.
> - We now also add a new baseline method in our comparisons, MMGPVAE, which is another very recent instance of dynamical VAEs that models latent factor distribution via Gaussian process and is also trained based on the ELBO optimization.
> - We also provide references to other variants of dynamical VAEs throughout our work both in the introduction and related work, such as Archer et al., 2015; Gao et al., 2016; Hurwitz et al., 2021 and Brenner et al., 2022. We further have a discussion in Section B.5 on the differences in optimization objectives across mmPLRNN and MRINE that are potential reasons for performance gains of MRINE over mmPLRNN. We hope that this is more clear now.
>
> &nbsp;
>
> > [W4] The equations in Section 2.1 contain inaccuracies, particularly in the notation used to describe multiscale dynamics ...
>
> We thank the reviewer for their careful examination of our work. We now use a different time index $t’$ for the Gaussian signal throughout the text. To summarize, we use time index $t$ for $s_t$ such that $t \in \mathcal{T}$; we then use another time index $t’$ for $y_{t’}$ such that $t’ \in \mathcal{T}’$. If $y_t$ is missing for a time step $t$, then that means that $t \notin \mathcal{T}’$, and so the modality-specific embedding and latent factors for $y_t$ are inferred by the internal dynamics of its modality-specific LDM (Eq. 7).

---

> ### Author Response · Authors · 2024-11-21
>
> > [W5] (Equation 8): The absence of a derivation for the loss function weakens the rigor of the methodology section ...
>
> We thank the reviewer for their comment. The loss objective in Eq. 8 is a negative k-step-ahead log-likelihood. Log-likelihood (or reconstruction for Gaussian log-likelihood) objectives are very commonly used to learn model parameters for time-series data given the nice theoretical properties of maximum-likelihood parameter estimation (Gilpin, 2020; Liu et al., 2023; Eldele et al., 2024).
>
> We aim to optimize future data log-likelihood as $L(\theta) = \sum_{i=1}^N  (\sum_{\substack{t \in \mathcal{T} \\ t \geq k}} \log  p_\theta(s_t) +  \sum_{\substack{t’ \in \mathcal{T’} \\ t’ \geq k}} \log  p_\theta(y_{t’}))$ where $\theta$ denotes MRINE parameters. Similar to many other autoencoder-based frameworks, the data log-likelihoods are modeled through latent factors $a_{t}$, such that k-step-ahead likelihood are $p_\theta(s_t) = Poisson(f_{\theta_s}(a_{1:t-k}))$ and $p_\theta(y_t’) = \mathcal{N}(f_{\theta_y}(a_{1:t’-k}, I)$ where $a_{1:t-k} = f_\phi (s_{1:t-k}, y_{1:t-k})$, where $\phi$ denotes MRINE's multiscale encoder.
>
> Of note, unlike VAE literature, this objective does not impose a structure on the posterior distribution of latent factors ($p_\phi(a_t|s_t, y_{t’})$) through a KL divergence term. Therefore, we can say that MRINE’s k-step-ahead prediction objective is similar to that of autoencoder reconstruction loss. Since we are not approximating any intractable integrals or imposing a probabilistic structure on the approximate posterior distribution ($p_\phi(a_t|s_t, y_{t’})$), there is no need for derivation of a variational lower bound, unlike VAE-based architectures that often include specific ELBO derivations.
>
> We provide details of our MRINE’s computation graph starting from observations to predicted likelihood parameters throughout Section 2.
> - Specifically, multiscale encoder design in Section 2.2 explains how to obtain k-step-ahead predicted or smoothed multiscale embedding and latent factors in detail, and how to account for missing samples via modality-specific LDMs.
> - Also, as explained in L220-221, the likelihood distribution parameters are obtained via forward-passing multiscale embedding factors through modality-specific decoder networks. To make these details easier to follow, we will add a summary of these steps to these sections. Providing this summary and the details regarding MRINE’s computation graph in Sections 2.2 and 2.3 would help readers reproduce our proposed approach. We also provided a link to our implementation in the footnote of the first page to facilitate the reproduction of our work.
>
> &nbsp;
>
>
> > [W6] The paper lacks a comprehensive comparison with state-of-the-art models in time-series decoding ...
>
> We thank the reviewer for their suggestion we would like to clarify two major points:
> - First, we agree that Mamba and S4 are important models in time-series decoding, but the novelty of MRINE lies in its ability to handle multimodal neural data with different timescale and missing samples. Mamba and S4 currently do not address these multimodal, multi-time-scale, or missing sample modeling challenges.
> - Second, and critically, we have compared with multiple very recent state-of-the-art baselines for models of multimodal neural signals, including mmPLRNN (Kramer et al., 2022), LFADS (Pandarinath et al., 2018), our multimodal extension of LFADS (mmLFADS) and now with MMGPVAE by Gondur et al., 2023. Within neuroscience, these are well-established and state-of-the-art methods for multimodal neural modeling (see publication dates and venues). So our baselines are not chosen selectively. In addition to adding MMGPVAE, by Gondur et al., 2023 in our comparisons now, we also now provide these comparisons in an entirely new dataset (Table 6 and GR Table 3 (or Table 7 in the revised manuscript)).
>
> &nbsp;
>
> > [W7] Figure 1 does not effectively illustrate the multiscale time dynamics central to MRINE’s approach, diminishing its ability to convey the model’s core features.
>
> We thank the reviewer for bringing this to our attention, this is a great point. We now updated both Fig. 1 and Fig. 2 as well as their captions to illustrate multiscale time dynamics more effectively by adding another timestep where $s_t$ is observed but $y_t$ is missing. Further, we would like to clarify that Fig. 1 is an overall illustration of MRINE’s model architecture, and multiscale time dynamics are the focus in Fig. 2.

---

> ### Author Response · Authors · 2024-11-21
>
> > [W8] The bar plots throughout the paper lack statistical tests and error bars, raising questions about the reliability ...
>
> The error bars were included in our bar plots in Fig. 4 and now Fig. 7, which are the only figures that contain bar plots. We apologize that these were not thick enough to easily see, and will increase the line thickness to improve their visibility. Also, our statistical significance results and p-values were given in the text itself, but to make it easier to see, we now added statistical significance levels using asterisks on error bars in the figures as well and further explained them in the captions. For reference, we provided statistical significance levels throughout the text for all analyses, for example in L319, 323, 360-379, 449, 1353-1354, 1357-1359, 1364-1365, 1611-1612.
>
> &nbsp;
>
> > [W9] While these techniques are positive additions for handling missing data, they require further explanation ...
>
> We agree with the reviewer that demonstrating such capabilities requires comparisons and ablations studies. We clarify that the following comparisons and ablations studies in the manuscript showcase these capabilities and we have now expanded them by adding baselines:
> -  In an ablation study in Appendix E.2, we trained MRINE models without specific loss terms in Eq. 9 (smoothed reconstruction) and Eq. 10 (smoothness regularization on $s_t$, $y_t$ and $x_t$) and showed that each term in Eq. 9 and Eq. 10 contributed to MRINE’s improved performance.
> - In Table 10 in Appendix E.4 (also shown in our global response GR Table 1), we performed a critical ablation study that shows the importance of MRINE’s multiscale encoder design for handling different time-scales as MRINE outperforms all baseline methods even when we add imputation and/or masking to them to train them with signals of different timescales. Similar results held in ablations in the new dataset (Table 7, also shown in our global response GR Table 3).
> - In addition, we now include multiple other baselines in our missing sample analysis in Fig. 6 and show that MRINE outperforms all baseline methods across all missing sample regimes.
>
> &nbsp;
>
> > [Q1] Could you clarify the novelty of MRINE relative to existing multimodal neural data models? ...
>
> Compared to all state-of-the-art deep learning baselines we compared to, the main novelty of MRINE lies in its **1)** ability to perform **real-time inference** while these methods perform non-causal inference, **2)** ability to address **different time-scales** and missing samples through its **novel multiscale encoder design**. This multiscale encoder design does not rely on any data augmentation for missing samples, e.g., zero-imputation as commonly used; instead, MRINE aims to learn modality-specific dynamical models to infer latent factors of the slower modality at time-points when its observations are missing, then aggregate information across the inferred latents from the modalities; this allows it to maintain the ability to infer latent factors in the faster modality’s timescale and to aggregate information across modalities.
>
> We show the impact of MRINE’s multiscale encoder design in the ablation study in Table 10 in Appendix E.4 (also provided in GR Table 1) and Table 7 (also provided in  GR Table 3 for the new dataset). In [W1] above, we detailed the differences between MRINE and Rezaei et al., 2023 and Coleman et al. 2011 (which are not deep learning models). Briefly, both Rezaei et al., 2023  and Coleman et al., 2011 are generalized linear multimodal models, similar to MSID that we compared to and thus do not capture nonlinearity. Also, neither of them addresses different timescales. Further, across all mentioned models, the training objectives of MRINE are distinct, which is another factor contributing to its improved performance as shown in Appendix E.2.
>
> &nbsp;
>
> > [Q2] Given that MRINE bears similarities to dynamical VAEs, could you discuss how MRINE’s approach compares to these methods, particularly regarding handling multiscale dynamics?
>
> Please see our response to the previous [W2] and [W3]. Further, please see our multiscale encoder design in Section 2.2 for details. We have compared MRINE to several state-of-the-art dynamical VAEs in neuroscience including mmPLRNN (Kramer et al., 2022), MMGPVAE (Gondur et al., 2023), LFADS (Pandarinath et al., 2018), mmLFADS (our extension over LFADS) showing the advantage of MRINE in handling different time-scales and missing samples.

---

> ### Author Response · Authors · 2024-11-21
>
> > [Q3] The log-likelihood calculation in Figure 3.C is not clearly explained ...
>
> We are not entirely sure what the reviewer is referring to as there is no log-likelihood calculation presented in our figures and Fig. 3 does not contain a panel C. This might have been a miscommunication and we are happy to provide further clarification.
>
> To compute k-step-ahead data log-likelihoods, we first need to infer the parameters of the likelihood distribution, which are obtained by forward passing $a_{t|t-k}$ through modality-specific decoder networks as we explained in L220-221. k-step-ahead multiscale embedding factors $a_{t|t-k}$ are obtained by multiplying k-step-ahead predicted multiscale latent factors $x_{t|t-k}$ (see L220-221) with the emission matrix $C$, where $x_{t|t-k}$ is obtained by forward predicting $x_{t-k|t-k}$ as $x_{t|t-k} = A^k x_{t-k|t-k}$ as explained in L217-218. $x_{t-k|t-k}$ is obtained by running Kalman filtering on the LDM in Eq. 1 and 2, whose inputs are multiscale encoder outputs $a_{1:t-k}$ as detailed in L215-216 of Section 2.2. The details on obtaining $a_t$'s through the multiscale encoder are explained throughout L172-214 in Section 2.2.
>
> After obtaining multiscale embedding factors, the log-likelihood for Poisson modality is computed as $\log(p_{\theta_s}(s_t | a_{t|t-k})) = -\lambda(a_{t|t-k}) + s_t \log(\lambda(a_{t|t-k})) - \log(s_t !)$ where $\lambda(a_{t|t-k})$ is the output of the Poisson decoder network parametrized by $\theta_s$.
>
> Similarly, for log-likelihood for Gaussian modality is computed as $\log(p_{\theta_y}(y_t | a_{t|t-k})) = \frac{-(y_t - \mu(a_{t|t-k}))^2}{2\sigma^2} - \frac{1}{2}\log(2\pi\sigma^2)$ where $\mu(a_{t|t-k})$ is the output of Gaussian decoder network parametrized by $\theta_y$. Note that $\sigma$ is set to a constant value instead of learning it from the data, as it yielded more stable performance (as explained in L161-162).
>
> As to how the log-likelihood relates to model performance, all loss terms in MRINE’s training objective rely on these log-likelihood calculations. As such, the log-likelihoods are a crucial part of parameter learning.
>
> &nbsp;
>
> Finally, we see that the reviewer raised the **flag for ethics review**, but we want to emphasize that we do not use any human data or any private dataset in this study. We also did not conduct any experiments on human subjects. All datasets used in this study are publicly available datasets often used in neuroscience for comparison purposes, for which we provide the references. As the original works that generated these datasets mention, all experiments were approved by the Institutional Animal Care and Use Committees of the institution at which these data were originally recorded.
>
> &nbsp;

---

> ### Author Response · Authors · 2024-11-21
>
> References:
>
> P. Ahmadipour, O. G. Sani, B. Pesaran, and M. M. Shanechi, “Multimodal subspace identification for modeling discrete-continuous spiking and field potential population activity,” J. Neural Eng., 2023, doi: 10.1088/1741-2552/ad1053.
>
> D. Kramer, P. L. Bommer, C. Tombolini, G. Koppe, and D. Durstewitz, “Reconstructing Nonlinear Dynamical Systems from Multi-Modal Time Series,” in Proceedings of the 39th International Conference on Machine Learning, PMLR, Jun. 2022, pp. 11613–11633. Available: https://proceedings.mlr.press/v162/kramer22a.html
>
> R. Gondur, U. B. Sikandar, E. Schaffer, M. C. Aoi, and S. L. Keeley, “Multi-modal Gaussian Process Variational Autoencoders for Neural and Behavioral Data,” presented at the The Twelfth International Conference on Learning Representations, Oct. 2023.  Available: https://openreview.net/forum?id=aGH43rjoe4
>
> M. R. Rezaei et al., “Inferring cognitive state underlying conflict choices in verbal Stroop task using heterogeneous input discriminative-generative decoder model,” J. Neural Eng., vol. 20, no. 5, p. 056016, Sep. 2023, doi: 10.1088/1741-2552/ace932.
>
> T. P. Coleman, M. Yanike, W. A. Suzuki, and E. N. Brown, “A Mixed-Filter Algorithm for Dynamically Tracking Learning from Multiple Behavioral and Neurophysiological Measures,” in The Dynamic Brain: An Exploration of Neuronal Variability and Its Functional Significance, P. Ding Mingzhou and P. Glanzman Dennis, Eds., Oxford University Press, 2011, p. 0. doi: 10.1093/acprof:oso/9780195393798.003.0001.
>
> H. Abbaspourazad, M. Choudhury, Y. T. Wong, B. Pesaran, and M. M. Shanechi, “Multiscale low-dimensional motor cortical state dynamics predict naturalistic reach-and-grasp behavior,” Nat Commun, vol. 12, no. 1, Art. no. 1, Jan. 2021, doi: 10.1038/s41467-020-20197-x.
>
> C. A. García, P. Félix, J. M. Presedo, and A. Otero, “Stochastic embeddings of dynamical phenomena through variational autoencoders,” Journal of Computational Physics, vol. 454, p. 110970, Apr. 2022, doi: 10.1016/j.jcp.2022.110970.
>
> E. Archer, I. M. Park, L. Buesing, J. Cunningham, and L. Paninski, “Black box variational inference for state space models,” Nov. 23, 2015, arXiv: arXiv:1511.07367. doi: 10.48550/arXiv.1511.07367.
>
> Y. Gao, E. W. Archer, L. Paninski, and J. P. Cunningham, “Linear dynamical neural population models through nonlinear embeddings,” in Advances in Neural Information Processing Systems, Curran Associates, Inc., 2016. Available: https://papers.nips.cc/paper_files/paper/2016/hash/76dc611d6ebaafc66cc0879c71b5db5c-Abstract.html
>
> C. L. Hurwitz et al., “Targeted Neural Dynamical Modeling,” presented at the Advances in Neural Information Processing Systems, Nov. 2021. Available: https://openreview.net/forum?id=dnDkuSzNh8
>
> M. Brenner, G. Koppe, and D. Durstewitz, “Multimodal Teacher Forcing for Reconstructing Nonlinear Dynamical Systems,” Dec. 15, 2022, arXiv: arXiv:2212.07892. Available: http://arxiv.org/abs/2212.07892
>
> W. Gilpin, “Deep reconstruction of strange attractors from time series,” in Advances in Neural Information Processing Systems, Curran Associates, Inc., 2020, pp. 204–216. Available: https://proceedings.neurips.cc/paper/2020/hash/021bbc7ee20b71134d53e20206bd6feb-Abstract.html
>
> Y. Liu, H. Zhang, C. Li, X. Huang, J. Wang, and M. Long, “Timer: Transformers for Time Series Analysis at Scale,” Feb. 04, 2024, arXiv: arXiv:2402.02368. doi: 10.48550/arXiv.2402.02368.
>
> E. Eldele, M. Ragab, Z. Chen, M. Wu, and X. Li, “TSLANet: Rethinking Transformers for Time Series Representation Learning,” in Proceedings of the 41st International Conference on Machine Learning, PMLR, Jul. 2024, pp. 12409–12428. Available: https://proceedings.mlr.press/v235/eldele24a.html
>
> C. Pandarinath et al., “Inferring single-trial neural population dynamics using sequential auto-encoders,” Nat Methods, vol. 15, no. 10, pp. 805–815, Oct. 2018, doi: 10.1038/s41592-018-0109-9.

---

> > ### Comment · Reviewer_9Zgw · 2024-12-02
> >
> > Thank you to the authors for their response. While it addresses some of my concerns, I will keep my score unchanged.

---

### Official Review · Reviewer_swya · 2024-11-02

**Soundness:** 2
**Presentation:** 2
**Contribution:** 2
**Rating:** 3
**Confidence:** 4

**Summary:**

The authors propose MRINE,  a dynamical modeling method that nonlinearly aggregates information across multiple modalities with different distributions and distinct timescales,  and missing samples. In addition, it can support inference in real time. The authors apply MRINE to two datasets, one synthetic (stochastic Lorenz attractor simulations) and one publicly available neuroscience dataset (NHP dataset).

**Strengths:**

1) Possible real world applications: Possibility of real time inference support is a great addition to many other multimodal models in neuroscience. This could possibly have positive applications to brain-computer interface research.
2) Robustness: The ability of the model to account for differing sampling rates and missing data is potentially impactful as much neuroscientific data is far from perfect, so models that take this into consideration are useful for real life applications.
3) The presentation is clear.

**Weaknesses:**

- **Poor coverage of the literature**: Despite the discussion of the literature in the paper and comparisons to different models, I find the coverage of the literature to be at a very surface level. There was much work that was not cited in the paper that should be included to accurately show the status of the field now.  For example [2,3] are multi-modal models and [1] despite not being multimodal can handle missing data. As it is now, the paper doesn’t do justice to all the other models that can handle multi-modal data and thus draws an inaccurate/incomplete picture of multi-modal work in neuroscience. To improve this, I suggest incorporating the citations below to the ‘Multimodal Models in Neuroscience’ section in the paper as well as the introduction.
- **Comparisons**: Despite comparing the model to several different ones, I am not sure why it was not compared to some of the novel multimodal neuroscience models [2,3]. I understand there are already 5 comparisons but adding a comparison with more current models like the ones previously stated can be more refreshing, and useful,  and show how MRINE compares to more current models.
- **Limited experiments**: The application of the model was limited, I wanted a demonstration with at least 1 more real experimental data to really make the model stand out and show its utility. Without validation on multiple real-world neuroscience datasets, it is challenging to assess whether MRINE’s claims generalize across different settings.


**References**
1. Ramchandran, S., Tikhonov, G., Kujanpää, K., Koskinen, M., & Lähdesmäki, H. (2021, March). Longitudinal variational autoencoder. In International Conference on Artificial Intelligence and Statistics (pp. 3898-3906). PMLR.
2. Gondur, R., Sikandar, U. B., Schaffer, E., Aoi, M. C., & Keeley, S. L. Multi-modal Gaussian Process Variational Autoencoders for Neural and Behavioral Data. In The Twelfth International Conference on Learning Representations.
3. Schneider, S., Lee, J. H., & Mathis, M. W. (2023). Learnable latent embeddings for joint behavioural and neural analysis. Nature, 617(7960), 360-368.
4. Zhou, D., & Wei, X. X. (2020). Learning identifiable and interpretable latent models of high-dimensional neural activity using pi-VAE. Advances in Neural Information Processing Systems, 33, 7234-7247.
5. Lee, M., & Pavlovic, V. (2021). Private-shared disentangled multimodal vae for learning of latent representations. In Proceedings of the ieee/cvf conference on computer vision and pattern recognition (pp. 1692-1700).

**Questions:**

1. Did you try to apply the model into more complex datasets where the behavior was not just 2d reaches? I understand this might be the standard kind of experimentation in BCI but I would be interested to learn about the behavioral decoding ability of MRINE with images or videos. In other words, how does MRINE handle high dimensional datasets?
2. How does the multiscale encoder in MRINE handle differences in noise characteristics between spike and LFP modalities + what impact does this have on the overall performance of behavior decoding?

---

> ### Author Response · Authors · 2024-11-21
>
> > [W1] Despite the discussion of the literature in the paper and comparisons to different models, I find the coverage of the literature to be at a very surface level ...
>
> - We thank the reviewer for their feedback. First, we should have been clearer about what we mean by multimodal. By multimodal model, we mean a model that takes multiple neural modalities as input (e.g., spikes, LFP) during **inference** to then predict a third modality (e.g., behavior). As such, our **inference** needs to aggregate information from **multiple** modalities.  There is a distinct category of models such as PSID (Sani et al., 2021), CEBRA (Schneider et al., 2023), or TNDM (Hurwitz et al., 2021) that take just one modality of neural data (e.g., spikes) as input during inference to predict a second modality (e.g., behavior). While these categories look at two modalities, they perform latent factor inference based on a single modality, i.e., neural data. We now cite these methods including CEBRA as recommended by the reviewer, and make this point clear in our introduction and related work.
>
> - We also now cite Lee et al., 2021 and apologize for missing it. This work is a multimodal variational autoencoder-based architecture utilizing products of experts, and we included citations to similar works such as MVAE by Wu & Goodman, 2018, MMVAE by Shi et al., 2019, MoPoE-VAE by Sutter et al., 2021 and MFM by Tsai et al., 2019. However, as we discussed in our related work, even though these works can handle time-series modalities with different timescales, they do not aggregate information across modalities at each time-step in real-time, rather, they process each modality with separate networks noncausally to encode each modality into a single vector (i.e., treating each time-series signal as a datapoint to be fused, rather than each time-step of corresponding time-series signals).
>
> - Despite being a unimodal model, we agree with the reviewer that Ramchandran et al., 2021 can handle missing data. We cite some unimodal models in related work that address missing samples such as linear dynamical models, and now add this reference there. Our literature review focused on multimodal models because addressing missing samples in the multimodal case is a distinct challenge as it needs to be handled while also allowing for multimodal fusion and as the time-steps of the missing samples for the different modalities may be different. We will expand there.
>
> - Further, we agree that MMGPVAE by Gondur et al., 2023 and pi-VAE by Zhou and Wei, 2020 are related to our work even if they are not designed to handle multiscale modalities (i.e., different timescales, which is the main focus of MRINE), and we now provide appropriate references to these works. As we detail below, we also now include MMGPVAE in our comparisons. We thank the reviewer for this excellent suggestion.
>
> > [W2] Despite comparing the model to several different ones, I am not sure why it was not compared to some of the novel multimodal neuroscience models ...
>
> We thank the reviewer for their great suggestion. We now include MMGPVAE in our comparisons in Table 1 (also shown in our global response, GR Table 2), and in our missing sample analysis (Fig. 6). We see that MRINE outperforms MMGPVAE, and this improvement is greater when modalities have different time-scales. We also emphasize that MMPGVAE is not designed to address the two major capabilities we designed MRINE for, which are **1)** enable real-time inference for multimodal data, and **2)** address different time-scales and missing samples. Indeed, MMGPVAE’s inference is non-causal. Further, even when we add imputation and/or masking to MMGPVAE, MRINE still significantly outperforms it for different time-scales, showing the advantage of its novel design for multiscale signals.  (Table 10 in Appendix E.4 and Table 7 in Appendix D.5 (also shown in GR Tables 1, 3).

---

> ### Author Response · Authors · 2024-11-21
>
> > [W3] The application of the model was limited, I wanted a demonstration with at least 1 more real experimental data ...
>
> We thank the reviewer for their suggestion, we agree that this is a great addition to our paper. We now include another public dataset from Flint et al., 2012 in our analysis, and we performed the same analysis as that in Fig. 4 as well as the baseline comparisons in Table 1 for this new dataset. The results can be found in Fig. 7, showing that MRINE can aggregate multimodal information. Furthermore, the baseline comparisons can be found in Table 6 (also shown below).
>
> | Model | 5 spike - 5 LFP (CC) | 10 spike - 10 LFP (CC) | 20 spike - 20 LFP (CC) |
> | -------- | ------- | ------- | ------- |
> | MSID | 0.444 &pm; 0.018 | 0.529 &pm; 0.021| 0.571 &pm; 0.021|
> | mmPLRNN    | 0.530 &pm; 0.022| 0.556 &pm; 0.025| 0.591 &pm; 0.027|
> | LFADS |0.443 &pm; 0.019| 0.450 &pm; 0.022| 0.414 &pm; 0.022|
> | mmLFADS | 0.395 &pm; 0.026| 0.443 &pm; 0.018| 0.519 &pm; 0.025|
> | MMGPVAE |0.558 &pm; 0.022| 0.624 &pm; 0.022| 0.670 &pm; 0.021|
> | MRINE | 0.550 &pm; 0.021| 0.628 &pm; 0.022| 0.664 &pm; 0.020|
> | MRINE - noncausal | 0.572 &pm; 0.022| 0.647 &pm; 0.023| 0.681 &pm; 0.021|
>
> For the same time-scale, MRINE outperforms all prior baselines, and also outperforms MMGPVAE when they both use the same neural data for inference ($p<0.006, n=15$, one-sided Wilcoxon-signed rank test; i.e., when similar to MMGPVAE, MRINE is allowed to do non-causal inference). Furthermore, for the same time-scale, even when MRINE just uses past neural data and thus less data than MMGPVAE, MRINE performs comparably to it ($p=0.11, n=15$, one-sided Wilcoxon-signed rank test; i.e., when MRINE does inference causally).
>
> Importantly, for the new dataset, again MRINE significantly outperforms MMGPVAE and mmPLRNN **when time-scales are different** (Table 7, or GR Table 3, and reiterated below for convenience).
>
> | Model | Behavior Decoding (CC) |
> | -------- | ------- |
> | MRINE | 0.661 &pm; 0.022|
> | mmPLRNN w/ Zero Imputation and Loss Masking | 0.538 &pm; 0.032|
> | MMGPVAE w/ Zero Imputation and Loss Masking |0.530 &pm; 0.030|
>
> This shows MRINE’s **multiscale modeling is crucial for modalities recorded with different timescales.**
>
> &nbsp;
>
> > [Q1] Did you try to apply the model into more complex datasets where the behavior was not just 2d reaches? ...
>
> There are two main reasons for our focus on multimodal neural datasets:
>
> -  First, as the reviewer stated, the main application focus of MRINE is on advancing brain-computer interfaces through information fusion across neural modalities. Please note that neural time-series data can be quite complex as they can have complex temporal structures and high noise. The goal of MRINE is to capture these complex spatiotemporal patterns in the presence of neural noise.
>
> -  We are not entirely sure what the reviewer means by “behavioral decoding ability of MRINE with images or videos”. If they mean images of the brain, these imaging modalities such as wide-field imaging are typically available during much simpler tasks than reaching as they are primarily done in rodents rather than NHPs. Further, we would need two simultaneous neural modalities to test MRINE, which is not typically done with imaging techniques. If the reviewer instead means just generic images and videos, not to do with neuroscience, this does not seem to be the use-case of MRINE as MRINE is designed to use multiple time-series modalities to predict a third time-series modality at every time-step. For standard video and text datasets used in multimodal NLP domain, this is generally not the case since 2 time-series modalities usually encode semantic information about a 3rd **static** modality, e.g., sentiment. In our related work, we gave references to studies that aim to develop multimodal frameworks for such scenarios.
>
> -  Third, the design choices of MRINE are **specifically made for neuroscience applications**: We use LDMs as our dynamical backbones so that **1)** MRINE can enable real-time inference in addition to noncausal inference, **2)** allow for modeling temporal stochasticity in the dynamics, **3)** allow for multiscale modeling of signals with different timescales. These real-time inference capabilities may not be as critical for generic video/image applications, unlike neuroscience applications.
>
> Finally, following the great suggestion by the reviewer, we now have added an entirely new dataset from a different lab, with different animals and a distinct task and show that the same conclusions hold in this new data, just showing the broad utility of MRINE for neuroscience and neurotechnology.

---

> ### Author Response · Authors · 2024-11-21
>
> > [Q2] How does the multiscale encoder in MRINE handle differences in noise characteristics between spike and LFP modalities ...
>
> The differences in noise characteristics between Poisson-distributed spiking signals and Gaussian-distributed LFP signals are accounted for in their corresponding modality-specific decoder networks. These decoders specifically learn the means of Poisson and Gaussian likelihoods for spike and LFP signals, respectively. These different likelihoods are in turn incorporated into the training loss, thus handling the different noise characteristics. As we show in our ablation study in Appendix E.3, modeling each modality with an appropriate distribution can help improve performance. Further, the smoothness regularizations for each modality are computed by using each modality’s specific observation distribution, which also helps MRINE to account for different noise characteristics of modalities (please also see the ablation study in Appendix E.2).
>
> &nbsp;
>
> References:
>
> O. G. Sani, H. Abbaspourazad, Y. T. Wong, B. Pesaran, and M. M. Shanechi, “Modeling behaviorally relevant neural dynamics enabled by preferential subspace identification,” Nat Neurosci, vol. 24, no. 1, pp. 140–149, Jan. 2021, doi: 10.1038/s41593-020-00733-0.
>
> S. Schneider, J. H. Lee, and M. W. Mathis, “Learnable latent embeddings for joint behavioral and neural analysis,” Nature, vol. 617, no. 7960, pp. 360–368, May 2023, doi: 10.1038/s41586-023-06031-6.
>
> C. L. Hurwitz et al., “Targeted Neural Dynamical Modeling,” presented at the Advances in Neural Information Processing Systems, Nov. 2021. Available: https://openreview.net/forum?id=dnDkuSzNh8
>
> M. Lee and V. Pavlovic, “Private-Shared Disentangled Multimodal VAE for Learning of Latent Representations,” presented at the Proceedings of the IEEE/CVF Conference on Computer Vision and Pattern Recognition, 2021, pp. 1692–1700. Available: https://openaccess.thecvf.com/content/CVPR2021W/MULA/html/Lee_Private-Shared_Disentangled_Multimodal_VAE_for_Learning_of_Latent_Representations_CVPRW_2021_paper.html
>
> Y. Shi, S. N, B. Paige, and P. Torr, “Variational Mixture-of-Experts Autoencoders for Multi-Modal Deep Generative Models,” in Advances in Neural Information Processing Systems, Curran Associates, Inc., 2019.  Available: https://proceedings.neurips.cc/paper/2019/hash/0ae775a8cb3b499ad1fca944e6f5c836-Abstract.html
>
> T. M. Sutter, I. Daunhawer, and J. E. Vogt, “Generalized Multimodal ELBO,” arXiv:2105.02470 [cs, stat], Jun. 2021, Available: http://arxiv.org/abs/2105.02470
>
> Y.-H. H. Tsai, P. P. Liang, A. Zadeh, L.-P. Morency, and R. Salakhutdinov, “Learning Factorized Multimodal Representations,” arXiv:1806.06176 [cs, stat], May 2019,  Available: http://arxiv.org/abs/1806.06176
>
> S. Ramchandran, G. Tikhonov, K. Kujanpää, M. Koskinen, and H. Lähdesmäki, “Longitudinal Variational Autoencoder,” in Proceedings of The 24th International Conference on Artificial Intelligence and Statistics, PMLR, Mar. 2021, pp. 3898–3906.  Available: https://proceedings.mlr.press/v130/ramchandran21b.html
>
> R. Gondur, U. B. Sikandar, E. Schaffer, M. C. Aoi, and S. L. Keeley, “Multi-modal Gaussian Process Variational Autoencoders for Neural and Behavioral Data,” presented at the Twelfth International Conference on Learning Representations, Oct. 2023. Available: https://openreview.net/forum?id=aGH43rjoe4
>
> D. Zhou and X.-X. Wei, “Learning identifiable and interpretable latent models of high-dimensional neural activity using pi-VAE,” in Advances in Neural Information Processing Systems, Curran Associates, Inc., 2020, pp. 7234–7247. Available: https://proceedings.neurips.cc/paper/2020/hash/510f2318f324cf07fce24c3a4b89c771-Abstract.html
>
> R. D. Flint, E. W. Lindberg, L. R. Jordan, L. E. Miller, and M. W. Slutzky, “Accurate decoding of reaching movements from field potentials in the absence of spikes,” J. Neural Eng., vol. 9, no. 4, p. 046006, Jun. 2012, doi: 10.1088/1741-2560/9/4/046006.

---

### Official Review · Reviewer_GtFZ · 2024-11-04

**Soundness:** 2
**Presentation:** 2
**Contribution:** 2
**Rating:** 3
**Confidence:** 3

**Summary:**

Summary: The paper presents a model named MRINE, which is designed to infer latent factors from multi-scale neural data in real time. The contribution of this model is claimed to generalize the latent variable models to multi-timescales, multi-probabilistic distributions, and allow missing samples across modalities. The model can work on discrete Poisson spiking activity and Gaussian local field potentials (LFPs). The evaluation is worked on simulated and real-world neural datasets, which show performance gain in subject behavior decoding and latent dynamics reconstruction.

**Strengths:**

1. MRINE's key components include a multiscale encoder for fusing modality-specific dynamics, a multiscale dynamical system backbone, and modality-specific decoders.
2. The writing and presentation of the paper is fairly good and easy for the readers to follow.
3. It's quite interesting for combing and aggregating the infromation from various modalites in neuroscience.

**Weaknesses:**

1. While I am inclined to hold the point that the quesiton/task this paper is focusing on has less scientific meaning, for model generalizaiton and for dealing with abnormal/missing data points.
2. From the modeling and ML perspective, the components in this MRINE's framework seems to be a combination of several existing techniques and empirical regularizers.

**Questions:**

1. What properties do the Poisson observations and Gaussian observations have in common?
2. A have no more questions, other concerns please refer to the Weakness section.

---

> ### Author Response · Authors · 2024-11-21
>
> We thank the reviewer for reviewing our manuscript and their comments.
>
> > [W1] While I am inclined to hold the point that the quesiton/task this paper is focusing on has less scientific meaning, for model generalizaiton and for dealing with abnormal/missing data points.
>
> -  As we discussed in our global response, the capabilities MRINE enables are critical for real-time neuroscience applications such as brain-computer interfaces (BCIs) and for studying multiscale neural processes (processes with different time-scales). This is because behavior is known to be encoded across different spatiotemporal scales of the brain that are measured with different modalities: spiking activity for small-scale neural processes at faster timescales and LFP for larger-scale network-level neural processes at slower timescales (Stavisky et al., 2015; Pesaran et al., 2018; Abbaspourazad et al., 2021; Ahmadipour et al., 2023). Indeed, these different modalities are measured with different time-scales. Thus, addressing different timescales is critical both for the scientific quest to study how the brain gives rise to behavior and for developing robust brain-computer interfaces (BCIs) that can aggregate information across modalities to better decode behavior and to be more robust to the loss of one modality (spikes) over time, which is a common problem in BCIs.
>
> In the global response, we now show that the innovations in MRINE including the multiscale encoder are critical to address different timescales and missing samples. Indeed, MRINE does so much more effectively than alternative baseline methods, even if these baselines use imputation and/or masking techniques. Please see the global response and GR Tables 1 and 3.
>
> -  In addition to time-scale differences, which are fundamental for studying the brain and BCIs, MRINE also addresses two other major problems: **i)** real-time and recursive inference causally in time (i.e., just using past neural samples), and **ii)** missing samples. Unlike MRINE, the inference in the multimodal baselines are non-causal. These capabilities are also critical for BCI technologies and real-time neuroscience applications as BCIs need to run in real-time and as missing samples is a common problem in BCIs, for example, due to wireless link interruption when transmitting data (Berger et al., 2020; Dastin-van Rijn et al., 2021; Gilron et al., 2021). Because MRINE can be quite beneficial in addressing these challenges in neuroscience applications and BCIs, we submitted it to the **applications to neuroscience track**.
>
> -  Finally, as we detailed in our introduction and methodology, every design element of MRINE is developed to reach these capabilities and is critical for these goals as is clear in our ablation studies including in  GR Table 1 and GR Table 3 in our global response. We expand on our design choices in the point below.

---

> ### Author Response · Authors · 2024-11-21
>
> > [W2] From the modeling and ML perspective, the components in this MRINE's framework seems to be a combination of several existing techniques and empirical regularizers.
>
> This is an important point to clarify. While the basic individual building blocks in MRINE (e.g., MLPs used as part of the encoder) exist, the novelty in MRINE is in how to use these basic blocks to develop a novel overall architecture, training method, and inference method that solves problems of importance to neuroscience and neurotechnology: **real-time multimodal inference, different timescales, missing samples**. We clarify that this is similar to many important methods in ML that use existing basic blocks to come up with a new method, for example, all of our baselines are similar in this respect.  Here are MRINE’s differences with existing techniques:
>    -    MRINE’s multiscale encoder architecture is novel and designed to account for different time-scales of different modalities (see also [W1] above) by using modality-specific LDMs to predict the value of samples at intermittent timesteps for the slower modality and MLP networks that then aggregate the resulting modality specific embedding factors (see Section 2.2 for design details). GR Table 1 and GR Table 2 show that this novel multiscale encoder design is critical for addressing different time-scales, and significantly outperforms baselines even when we add imputation and/or masking to them.
>   -  As we state in Section 2.3, smoothness regularization was first proposed by Li et al., 2021 for anomaly detection applications rather than for multimodal modeling, and further it was limited to Gaussian distributions. We emphasize that **1)** we extend this technique to Poisson distributed modalities, and doing so significantly contributes to our model’s performance (see Appendix E.2), and **2)** the application of smoothness regularization in our context is very different than that in Li et al., 2021.
>   -  Further, as we detail in Appendix A, the time-dropout in MRINE is distinct from masking in masked autoencoders. Here, time-dropout aims to improve the network’s robustness to missing samples rather than predicting missing samples from existing ones. In addition, our time-dropout randomly drops the **whole input signal for a given time-step to mimic missing sample scenarios**, which is conceptually distinct from randomly dropping individual elements of recurrent states to prevent overfitting in some prior work.
>
> We emphasize again that the main novelty of MRINE lies in designing a new technique for addressing problems in important applications in neuroscience, rather than proposing new individual deep learning components. Thus, MRINE was submitted under **applications to neuroscience track**. As such, to show the impact of MRINE on neuroscience applications, we perform extensive analysis on real monkey datasets – with a new dataset now added – and show that MRINE can
>
> 1. Successfully aggregate information across modalities (Fig. 4 and Fig. 7),
> 2. Do so casually in time, unlike prior multimodal models that have non-causal inference,
> 3. Achieves better behavior decoding performance than recent linear and nonlinear multimodal models, especially for different time-scales (Table 1, Table 6, Table 7 (GR Table 3) and Table 10 (GR Table 1)), and
> 4. Also robust to missing samples (Fig. 5) and outperforms baseline methods in missing sample scenarios (Fig. 6).
>
> > [Q1] What properties do the Poisson observations and Gaussian observations have in common?
>
> Both of these observations are made in the premotor cortex and encode information about the downstream behavior signal, which is the arm reaching kinematics. As we detail in Appendix D.1, both modalities are recorded by a 96-channel microelectrode array implanted in the premotor cortex of a non-human primate (NHP) while it performs a reaching task. Spike signals (Poisson modality) are discrete observations obtained from the raw signal (recorded by microelectrodes) by threshold-crossing, and LFP signals (Gaussian modality) are obtained by low-pass filtering the raw neural signals with 300 Hz cutoff frequency. In that sense, both modalities are recorded during the same task, but cover different frequency contents of neural activity: spiking activity captures the high-frequency content and LFP signals capture relatively slower frequencies. Similar statements are also valid for the new dataset that we added in our results, please see Appendix D.2 for details. As shown in prior work (Stavisky et al., 2015; Abbaspourazad et al., 2021; Ahmadipour et al., 2023), both signals can encode information about the downstream behavior signal.

---

> ### Author Response · Authors · 2024-11-21
>
> References:
>
> Stavisky, S. D., Kao, J. C., Nuyujukian, P., Ryu, S. I., & Shenoy, K. V. (2015). A high performing brain–machine interface driven by low-frequency local field potentials alone and together with spikes. Journal of Neural Engineering, 12(3), 036009. https://doi.org/10.1088/1741-2560/12/3/036009.
>
> B. Pesaran et al., “Investigating large-scale brain dynamics using field potential recordings: analysis and interpretation,” Nat Neurosci, vol. 21, no. 7, pp. 903–919, Jul. 2018, doi: 10.1038/s41593-018-0171-8.
>
> P. Ahmadipour, O. G. Sani, B. Pesaran, and M. M. Shanechi, “Multimodal subspace identification for modeling discrete-continuous spiking and field potential population activity,” J. Neural Eng., 2023, doi: 10.1088/1741-2552/ad1053.
>
> H. Abbaspourazad, M. Choudhury, Y. T. Wong, B. Pesaran, and M. M. Shanechi, “Multiscale low-dimensional motor cortical state dynamics predict naturalistic reach-and-grasp behavior,” Nat Commun, vol. 12, no. 1, Art. no. 1, Jan. 2021, doi: 10.1038/s41467-020-20197-x.
>
> M. Berger, N. S. Agha, and A. Gail, “Wireless recording from unrestrained monkeys reveals motor goal encoding beyond immediate reach in frontoparietal cortex,” eLife, vol. 9, p. e51322, May 2020, doi: 10.7554/eLife.51322.
>
> E. M. Dastin-van Rijn, N. R. Provenza, M. T. Harrison, and D. A. Borton, “How do packet losses affect measures of averaged neural signalsƒ,” Annu Int Conf IEEE Eng Med Biol Soc, vol. 2021, pp. 941–944, Nov. 2021, doi: 10.1109/EMBC46164.2021.9629666.
>
> L. Li, J. Yan, H. Wang, and Y. Jin, “Anomaly Detection of Time Series with Smoothness-Inducing Sequential Variational Auto-Encoder,” arXiv.org. Available: https://arxiv.org/abs/2102.01331v1.
>
> R. Gilron et al., “Long-term wireless streaming of neural recordings for circuit discovery and adaptive stimulation in individuals with Parkinson’s disease,” Nat Biotechnol, vol. 39, no. 9, pp. 1078–1085, Sep. 2021, doi: 10.1038/s41587-021-00897-5.

---

### Official Review · Reviewer_44y3 · 2024-11-04

**Soundness:** 3
**Presentation:** 2
**Contribution:** 2
**Rating:** 6
**Confidence:** 3

**Summary:**

This paper presents MRINE, a nonlinear dynamical modeling framework for multiscale neural activity that enables real-time information aggregation across modalities with different timescales and/or missing samples. MRINE consists of a multiscale encoder that fuses modalities after learning within-modality dynamics, a multiscale dynamical backbone for extracting multimodal temporal dynamics and enabling real-time decoding, and modality-specific decoders to account for different probabilistic distributions. The authors also introduce smoothness regularization objectives and a time-dropout technique to improve robustness to missing samples. Through simulations and application to a real multiscale brain dataset, MRINE is shown to outperform prior linear and nonlinear multimodal models in fusing information across modalities to improve decoding, even with different timescales or missing samples, making it valuable for real-time neurotechnology applications.

**Strengths:**

- Modeling neural dynamics across varying modalities.
- The ability to handle different timescales and missing samples.
- Enabling real-time decoding.

**Weaknesses:**

- My main concern is that it’s difficult to find significant scientific insights in this work. It feels more like a combination of various features to achieve an improvement in behavioral decoding accuracy over the baselines, yet the 61% accuracy does not seem particularly high. In other words, the improvement in decoding accuracy alone doesn’t convincingly justify the introduction of different timescales and handling of missing samples, as I think baseline accuracy could potentially be enhanced through additional parameters or optimized training settings, such as changing the factor dimension in LFADS.

**Questions:**

- Why were only 20 spike and LFP channels used? This seems like a small number. Does this mean the input feature dimension of the data is 20+20=40?
- Is it possible to compare the accuracy of baselines with missing samples in Section 3.3?

---

> ### Author Response · Authors · 2024-11-21
>
> > [W1] ... It feels more like a combination of various features to achieve an improvement in behavioral decoding accuracy over the baselines ...
>
> We thank the reviewer for their comment. We should clarify multiple points here.
>
> -  First, the reported metrics for behavior decoding are correlation coefficient (CC) and not percent correct, and thus baseline random chance CC is 0, unlike 50% for random prediction accuracy in classification tasks. Behavior decoding accuracies **for neural datasets such as ours in which trials are designed to mimic naturalistic settings (e.g., having variable length trials with less structured movements)**, can be in a similar range (Ahmadipour et al., 2023;  Sani et al., 2021; Flamary and Rakotomamonjy, 2012; Xie et al., 2018), especially when decoding is performed from the latent factors of unsupervised models. Furthermore, since absolute performance depends on task difficulty and quality of recording in a dataset and so is dataset-specific, what matters is the comparison across baselines. MRINE outperforms all these baselines, and this improvement is especially substantial when modalities have multiple time-scales (see global response, GR Table 1).
>
> -  Second, the goal of MRINE was not simply to “achieve an improvement in behavioral decoding accuracy over the baselines”. As we clarify in global response, the goal and novelties were to enable **1)** real-time causal inference unlike the baselines whose inference is non-causal, **2)** address modalities with different time-scales (multiscale signals) and missing samples. We now show that the innovations in MRINE (the multiscale encoder, etc) are critical to achieve these goals, and even adding imputation and/or masking to these baselines still significantly underperformed MRINE. (Table 6 and Table 10, also provided in our global response GR Table 1 and GR Table 3).
>
> To provide even further evidence, per the reviewer's request, we trained mmLFADS models on different timescale signals, and with 128-dimensional factors and doubled the RNN networks' capacity (hidden units). In contrast, every other model is trained with 64-dimensional latent states (except MMGPVAE due to shared latent structure, thus, behavior decoding with MMGPVAE is performed with 96-dimensional latent states). In this scenario, behavior decoding performance with mmLFADS increased to 0.385 &pm; 0.010 from 0.253 &pm; 0.023 (see GR Table 1), and it again underperformed MRINE, MMGPVAE and mmPLRNN. This result again shows that multiscale modeling is crucial for different timescale signals beyond hyperparameter choices.
>
> > [Q1] Why were only 20 spike and LFP channels used? This seems like a small number. Does this mean the input feature dimension of the data is 20+20=40?
>
> Yes, the reviewer is correct about the feature dimension being 40 (20 spiking channels and 20 LFP channels) for the original NHP dataset. We now add a new dataset in which the feature dimensions are picked at 120 (20 spiking channels and 5 power bands from 20 LFP channels, resulting in 100-dimensional inputs for LFP power signals) and we show similar results.
>
> We also thank the reviewer for raising the question about the choice of the number of features, which is important to clarify. There are two reasons we focused on 20 channels.
>
> - First, the number of neurons it takes for decoding performance to saturate depends on the task complexity and this number is typically around or less than 20 neurons for benchmark neural datasets including the public data we had access to. For this reason, we went up to 20 neurons because in that regime performance was not saturated so we could ask whether multimodal methods (MSID, mmPLRN, MRINE, mmLFADS, and now MMGPVAE) can aggregate information across modalities to improve performance. Otherwise, decoding would have been saturated with the first modality so we could not test this capability for any of the methods, MRINE, or the alternatives. This saturation is also observed in the prior study that tested MSID for example.
> -  Second, while in laboratory neuroscience experiments one can record more than 20 neurons these days, this is not the case for chronic brain-computer interfaces (BCIs) because they are based on **implantable devices with fewer channels** and further because **they lose the signal over time due to scar tissue formation**. Indeed, one main advantage of multimodal aggregation would be in low to mid-information regimes for one modality, such that the other modality can help improve accuracy and robustness as also shown in prior studies (Bansal et al., 2012). For example, prior studies show that spiking activity from implanted electrodes may degrade faster than local field potentials (Stavisky et al., 2015) leading to a small number of spike channels. This is exactly the scenario where using more robust LFP signals is the most advantageous as it can provide significant robustness and performance improvements as we show in our case study.

---

> ### Author Response · Authors · 2024-11-21
>
> > [Q2] Is it possible to compare the accuracy of baselines with missing samples in Section 3.3?
>
> We thank the reviewer for their suggestion. We now extend our missing sample analysis to also include MSID, mmPLRNN, MMGPVAE, LFADS, and mmLFADS. As we show in Fig. 6 of our revised manuscript, MRINE outperforms all baseline methods in behavior decoding in the presence of missing samples.
>
> &nbsp;
>
> References:
>
> R. Flamary and A. Rakotomamonjy, “Decoding Finger Movements from ECoG Signals Using Switching Linear Models,” Front. Neurosci., vol. 6, Mar. 2012, doi: 10.3389/fnins.2012.00029.
>
> Z. Xie, O. Schwartz, and A. Prasad, “Decoding of finger trajectory from ECoG using deep learning,” J. Neural Eng., vol. 15, no. 3, p. 036009, Feb. 2018, doi: 10.1088/1741-2552/aa9dbe.
>
> O. G. Sani, H. Abbaspourazad, Y. T. Wong, B. Pesaran, and M. M. Shanechi, “Modeling behaviorally relevant neural dynamics enabled by preferential subspace identification,” Nat Neurosci, vol. 24, no. 1, Art. no. 1, Jan. 2021, doi: 10.1038/s41593-020-00733-0.
>
> P. Ahmadipour, O. G. Sani, B. Pesaran, and M. M. Shanechi, “Multimodal subspace identification for modeling discrete-continuous spiking and field potential population activity,” J. Neural Eng., 2023, doi: 10.1088/1741-2552/ad1053.
>
> Bansal, A. K., Truccolo, W., Vargas-Irwin, C. E., & Donoghue, J. P. (2012). Decoding 3D reach and grasp from hybrid signals in motor and premotor cortices: Spikes, multiunit activity, and local field potentials. Journal of Neurophysiology, 107(5), 1337–1355. https://doi.org/10.1152/jn.00781.2011.
>
> Stavisky, S. D., Kao, J. C., Nuyujukian, P., Ryu, S. I., & Shenoy, K. V. (2015). A high performing brain–machine interface driven by low-frequency local field potentials alone and together with spikes. Journal of Neural Engineering, 12(3), 036009. https://doi.org/10.1088/1741-2560/12/3/036009.

---

### Author Response · Authors · 2024-11-21
**Global Response - 1**

We thank the reviewers for their comments and suggestions on our work. We are pleased that the reviewers find that our problem formulation on combining and aggregating information across neural data modalities is ‘quite interesting’ (reviewer GtFZ), ‘possibility of real time inference support’ is a ‘great addition to many other multimodal models in neuroscience’ which can have ‘positive applications to brain-computer interface research’ (reviewer swya) and our ‘efforts to handle missing data with techniques like smoothness regularization and time-dropout are noteworthy additions’ (reviewer 9Zgw).

Based on the comments, however, it seems that our main contributions may have been lost in the discussions about performance alone. Thus, before presenting our new results, we would like to emphasize that **the main novelty of MRINE lies in providing two main capabilities for multimodal neural modeling, rather than simply improving performance**:

1.  Enabling recursive real-time nonlinear inference, which is critical for closed-loop ML/AI systems such as brain-computer interfaces. In contrast, most deep learning models of neural data – including our multimodal baselines and the new models that reviewers asked for comparison to – only perform non-causal non-recursive inference.
2.  Modeling multiscale neural signals by accounting for their time-scale differences and missing samples, rather than pure performance gains when all samples are available. These are critical capabilities for real-time applications such as brain-computer interfaces (see also Introduction and Related Work).

Therefore, we would like to highlight the ablation study we performed in Appendix E.4 (which now includes a new baseline we added), which may have been overlooked as we provided this result in our Appendix. In this ablation study, we show that even if the alternative models perform global mean-imputation (i.e., zero-imputation due to z-scoring) and/or masking for the missing LFP samples in their training objective, MRINE still significantly outperforms them.

**GR Table 1**:
| Model    | Behavior Decoding (CC) |
| -------- | ------- |
| MRINE | 0.611 &pm; 0.012    |
| MRINE w/ Zero Imputation | 0.523 &pm; 0.013 |
| MRINE w/ Zero Imputation and Loss Masking | 0.581 &pm; 0.014    |
| mmPLRNN w/ Zero Imputation| 0.498 &pm; 0.009    |
| mmPLRNN w/ Zero Imputation and Loss Masking |0.539 &pm; 0.011     |
| mmLFADS w/ Zero Imputation | 0.280 &pm; 0.022   |
| mmLFADS w/ Zero Imputation and Loss Masking |0.253 &pm; 0.023    |
| MMGPVAE w/ Zero Imputation | 0.393 &pm; 0.023     |
| MMGPVAE w/ Zero Imputation and Loss Masking | 0.518 &pm; 0.011    |

As expected, all models have improved performance when missing samples are masked from the loss function (except mmLFADS which has poor performance due to high distortion in initial condition summarization), but regardless, MRINE outperforms all models including its own variants that add imputation and/or masking in training. These results clearly show that even adding imputation and/or masking are significantly inferior to the innovations in MRINE’s multiscale encoder to address multiple timescales. Thus, MRINE’s **multiscale modeling is crucial for modalities recorded with different timescales. This clearly shows the advantage of new capability 2 above.**  The new capability 1 is extensively shown in Figures 5 and 6.

---

> ### Author Response · Authors · 2024-11-21
> **Global Response - 2**
>
> Now, we summarize the main new analyses/results per reviewers’ requests. Our responses will refer to the uploaded revised version of our manuscript for new results, tables, figures, and line numbers.
>
> -   **New analysis 1: Adding a recent multimodal baseline** (MMGPVAE): **While MRINE’s main novelty is to enable the above two capabilities, which are not achieved by prior models of neural activity**, we found that MRINE also improves performance over prior multimodal models of neural activity. Per requests of reviewers swya and 9Zgw, we now added another method, namely MMGPVAE  by Gondur et al., 2023 to our baselines in Table 1. We emphasize the MMGPVAE’s inference is non-causal and thus uses both past and future neural data, unlike the causal/real-time inference in MRINE. GR Table 1 above showed this comparison when modalities had different time-scales, finding that MRINE significantly outperforms MMGPVAE. The below table shows the new comparison with modalities having the same timescales. MRINE also outperforms MMGPVAE even when time-scales are the same, where MMGPVAE is now the most competitive baseline for the mid and high information regimes, i.e., 10 or 20 channels:
>
> **GR Table 2**:
> | Model | 5 spike - 5 LFP (CC) | 10 spike - 10 LFP (CC) | 20 spike - 20 LFP (CC) |
> | -------- | ------- | ------- | ------- |
> | MVAE | 0.326 &pm; 0.011 | 0.386 &pm; 0.009 | 0.425 &pm; 0.009 |
> | MSID | 0.380 &pm; 0.021 | 0.440 &pm; 0.015 | 0.519 &pm; 0.012 |
> | mmPLRNN    | 0.455 &pm; 0.012| 0.478 &pm; 0.011| 0.533 &pm; 0.012|
> | LFADS |0.467 &pm; 0.017| 0.495 &pm; 0.015| 0.548 &pm; 0.011|
> | mmLFADS | 0.468 &pm; 0.016| 0.507 &pm; 0.015| 0.547 &pm; 0.011|
> | MMGPVAE |0.424 &pm; 0.012| 0.511&pm; 0.014| 0.579 &pm; 0.009 |
> | MRINE | 0.487 &pm; 0.007| 0.555 &pm; 0.011 | 0.611 &pm; 0.012|
> | MRINE - noncausal | 0.519 &pm; 0.009| 0.573 &pm; 0.011| 0.621 &pm; 0.011|
> In addition, we also added MMGPVAE in:
> * The visualizations inferred latent trajectories in **Fig. 9** (previously Fig. 8)
> * Our ablation study on the importance of multiscale encoder design is in Table 10 (previously Table 7) in Appendix E.4 (also provided above in GR Table 1 for convenience).
>
> &nbsp;
>
>
> -  **New Analysis 2: Adding a new dataset**: As per reviewer swya’s request, we now added another neural dataset by Flint et al., 2012 to our analyses. In this dataset, spiking activity and LFP signals were recorded during a center-out reaching task, and we modeled 10 ms spiking signals and 10 ms or 50 ms LFP power signals extracted in 5 different power bands for each LFP recording channel. **We present the results for this new dataset in Appendix D.5** of our revised manuscript. Results were consistent with the original dataset, with MRINE significantly outperforming all baselines. We performed the same analysis as in Fig. 4 where we fused various combinations of spiking and LFP channels and reported behavior decoding accuracies (Fig. 7).  Similar results hold with MRINE performing more accurate inference than MMGPVAE and mmPLRNN both when modalities have different timescales (where mmPLRNN and MMGPVAE are trained with imputed missing samples, see below GR Table) and even when the modalities have the same time-scales (see Table 6 in paper).
>
> **GR Table 3: Results on the entirely new dataset and with the new MMGPVAE baseline when modalities have different timescales**
> | Model | Behavior Decoding (CC) |
> | -------- | ------- |
> | MRINE | 0.661 &pm; 0.022 |
> | mmPLRNN w/ Zero Imputation and Loss Masking | 0.538 &pm; 0.032 |
> | MMGPVAE w/ Zero Imputation and Loss Masking | 0.530 &pm; 0.030 |
>
> &nbsp;
>
> -  **New Analysis 3: Adding other baseline methods to missing sample analysis in Section 2.3**: Per requests of reviewers 44y3 and 9Zgw, we now added all dynamical baseline models to our missing sample analysis in Section 2.3. In our initial version, we had the missing sample analysis only for MRINE in **Fig. 6 in Appendix D.3**, and now we updated this figure by including MSID, mmPLRNN, LFADS, mmLFADS, and also MMGPVAE. We show that MRINE outperforms the baseline methods across all missing sample ratios.
>
> References:
>
> R. Gondur, U. B. Sikandar, E. Schaffer, M. C. Aoi, and S. L. Keeley, “Multi-modal Gaussian Process Variational Autoencoders for Neural and Behavioral Data,” presented at the Twelfth International Conference on Learning Representations, Oct. 2023. Available: https://openreview.net/forum?id=aGH43rjoe4
>
> R. D. Flint, E. W. Lindberg, L. R. Jordan, L. E. Miller, and M. W. Slutzky, “Accurate decoding of reaching movements from field potentials in the absence of spikes,” J. Neural Eng., vol. 9, no. 4, p. 046006, Jun. 2012, doi: 10.1088/1741-2560/9/4/046006.

---

### Meta-Review · Area_Chair_BFDr · 2024-12-19

**Metareview:**

This paper introduces MRINE, a framework for real-time decoding of neural activity across multiple modalities, addressing challenges like missing samples and different timescales. It demonstrates improved decoding performance using both simulated and real-world datasets, with applications in neurotechnology.

Strengths include effective handling of missing data through techniques like smoothness regularization and time-dropout. The presentation is clear, and the real-time decoding capability is a valuable addition to multimodal neural data processing, with potential applications in brain-computer interfaces.

However, the scientific contribution of the paper is unclear, as the reported decoding accuracy (61%) is not significant enough to justify the complexity of the proposed model. Additionally, the paper lacks sufficient comparison with state-of-the-art models in multimodal neural data processing, which makes it difficult to evaluate MRINE's uniqueness and effectiveness. The theoretical foundations, particularly regarding multiscale dynamics, are weak, and the methodology lacks rigorous mathematical derivations, compromising reproducibility. Furthermore, the absence of error bars in performance visualizations raises concerns about the reliability of the reported results. These weaknesses undermine the paper’s scientific impact and rigor, leading to the recommendation for rejection.

**Additional Comments On Reviewer Discussion:**

During the rebuttal period, the authors made revisions addressing reviewers' concerns, but key issues remained unresolved. They added comparisons to dynamical VAEs like LFADS and MMGPVAE, but reviewers still felt MRINE's novelty was not fully established. The authors updated equations and provided new data on handling missing samples, yet the reviewers felt the explanations were insufficient for managing multimodal data effectively. Importantly, the reviewers raised concerns about the novelty of MRINE, suggesting that it seemed more like an incremental improvement over existing methods, particularly due to its reliance on empirical regularizers and well-established techniques. While the authors emphasized MRINE's unique architecture and real-time inference capabilities, the reviewers felt that these features did not offer enough of a breakthrough to distinguish it significantly from other approaches. In summary, while the authors made meaningful revisions and provided additional data, the reviewers remained unconvinced of MRINE’s novelty and the scientific value of its behavioral decoding task.

---

### Decision · Program_Chairs · 2025-01-22

Reject